# Fast Bellman Updates for Wasserstein Distributionally Robust MDPs

**Zhuodong Yu**[1]    **Ling Dai**[1]    **Shaohang Xu**[1]    **Siyang Gao**[2]    **Chin Pang Ho**[1]

[1]School of Data Science, City University of Hong Kong, Hong Kong
[2]Department of Systems Engineering, City University of Hong Kong, Hong Kong
{zhuodonyu2-c, lingdai5-c, shaohanxu2-c}@my.cityu.edu.hk
{siyangao, clint.ho}@cityu.edu.hk

## Abstract

Markov decision processes (MDPs) often suffer from the sensitivity issue under model ambiguity. In recent years, robust MDPs have emerged as an effective framework to overcome this challenge. Distributionally robust MDPs extend the robust MDP framework by incorporating distributional information of the uncertain model parameters to alleviate the conservative nature of robust MDPs. This paper proposes a computationally efficient solution framework for solving distributionally robust MDPs with Wasserstein ambiguity sets. By exploiting the specific problem structure, the proposed framework decomposes the optimization problems associated with distributionally robust Bellman updates into smaller subproblems, which can be solved efficiently. The overall complexity of the proposed algorithm is quasi-linear in both the numbers of states and actions when the distance metric of the Wasserstein distance is chosen to be $L_1$, $L_2$, or $L_\infty$ norm, and so the computational cost of distributional robustness is substantially reduced. Our numerical experiments demonstrate that the proposed algorithms outperform other state-of-the-art solution methods.

## 1 Introduction

Markov Decision Processes (MDPs) provide a flexible and powerful modeling framework for sequential decision-making problems under uncertainty (Puterman, 2014; Sutton and Barto, 2018). However, the standard MDP model assumes that the exact knowledge of the transition kernel is available, which is not the case for most real-world applications. While these model parameters can be estimated from data, it is well-known that the optimal policy of MDP is sensitive to estimation errors because of the dynamic nature of the problem. In particular, small errors in estimating transition kernels could lead to catastrophic failures in practice (Iyengar, 2005; Nilim and El Ghaoui, 2005; Le Tallec, 2007).

To overcome the aforementioned challenge in model ambiguity, robust MDPs assume that the transition kernels belong to ambiguity sets (Iyengar, 2005; Nilim and El Ghaoui, 2005; Xu and Mannor, 2006). Through optimizing the worst-case performance, robust MDPs compute robust policies to avoid disappointing performance due to the sensitivity issue caused by inaccurate transition kernels. However, robust MDPs are often too conservative because they always adopt the worst possible transition kernels without any consideration of their likelihood. To mitigate this undesirable side effect of robust MDPs, distributionally robust MDPs are proposed to incorporate distributional information of model ambiguity and maximize the expected reward under the most adversarial probability distribution (Xu and Mannor, 2010; Yu and Xu, 2015).

In a distributionally robust MDP, we assume that all the plausible distributions of the transition kernel belong to a prescribed ambiguity set, which could be formalized in different types, such as moment ambiguity sets (Delage and Ye, 2010) and Wasserstein ambiguity sets (Mohajerin Esfahani and

37th Conference on Neural Information Processing Systems (NeurIPS 2023).

Kuhn, 2018). In this paper, we focus on the Wasserstein ambiguity sets, which have been a popular choice for distributionally robust data-driven optimization in recent years because of their outstanding empirical performance as well as nice theoretical properties, such as consistency in optimality and finite-sample bounds (Mohajerin Esfahani and Kuhn, 2018; Gao and Kleywegt, 2022). By using Wasserstein distance to formulate the ambiguity set, Yang (2017) shows that there exists an optimal policy that is stationary and Markovian for the corresponding Wasserstein distributionally robust MDP.

While (distributionally) robust MDPs can be solved by extending standard solution methods in classical MDPs to their (distributionally) robust counterparts, these solution methods become much more computationally demanding. For example, each Bellman update for (distributionally) robust MDPs can be formulated as a convex optimization problem. Without making use of any specific problem structure, one would need to use generic convex optimization solvers to compute these Bellman updates, which have to be evaluated numerous times for computing the (distributionally) robust value function. This computational challenge restricted the application of (distributionally) robust MDPs to small or medium size of problems. In recent years, many efficient algorithms are proposed for solving robust MDPs to address this issue (Iyengar, 2005; Nilim and El Ghaoui, 2005; Ho et al., 2018; Behzadian et al., 2021; Grand-Clément and Kroer, 2021b; Ho et al., 2022).

On the other hand, however, distributionally robust MDPs have received limited attention on their computational efficiency. To the best of our knowledge, (Grand-Clément and Kroer, 2021a) is the only pioneer work that proposes a first-order method to efficiently solve Wasserstein distributionally robust MDPs. However, being restricted by the nature of first-order methods, the proposed algorithm in (Grand-Clément and Kroer, 2021a) struggles when high accuracy is needed or when the discount factor is close to one, as it would take unsatisfactorily many numbers of iterations to compute the distributionally robust value function.

In this paper, we take an alternative approach to develop fast algorithms for solving Wasserstein distributionally robust MDPs. In particular, we reformulate the optimization problem of the distributionally robust Bellman update to a structural form, and by exploiting the specific problem structure, we propose decomposition schemes in which the problem of interest can be decomposed into smaller subproblems, which can be solved by our customized algorithms in quasi-linear time. As we will show later, the time complexities of the proposed algorithms are linear in the numbers of actions and kernels and quasi-linear in the number of states, which are significantly better than the state-of-the-art solution methods (Xu and Mannor, 2010; Grand-Clément and Kroer, 2021a).

The rest of this paper is organized as follows. Section 2 introduces the related work, and Section 3 describes the setting of Wasserstein distributionally robust MDPs. In Section 4, we illustrate our decomposition scheme for computing the distributionally robust Bellman updates when the Wasserstein distance in the ambiguity set is chosen to be $L_q$ norm. In Section 5, customized fast algorithms and their time complexities are particularly provided for the three most well-known cases: $L_1$, $L_2$, and $L_\infty$. The numerical experiments in Section 6 verify that our methods outperform other state-of-the-art algorithms.

**Notations** We use boldface lowercase and uppercase letters to denote vectors and matrices, respectively. We denote by $[N]$ the set of natural numbers from 1 to $N$; that is, $[N] = \{1, 2, \ldots, N\}$. The vector $\boldsymbol{e}$ is denoted as the vector of all ones while the dimension depends on the context. The vector $\boldsymbol{e}_j$ represents the vector of all zeros except the $j$-th component is one. Probability simplex is denoted as $\Delta_N = \left\{ \boldsymbol{x} \in \mathbb{R}^N : \boldsymbol{e}^\top \boldsymbol{x} = 1, \boldsymbol{x} \geq \boldsymbol{0} \right\}$. The smallest component of a vector $\boldsymbol{x}$ is denoted by $\min\{\boldsymbol{x}\}$. Given a set $\mathcal{X}$, we call the set of all the Borel probability measures as $P(\mathcal{X})$. The general $L_q$ norm for a vector $\boldsymbol{x} \in \mathbb{R}^N$ is denoted by $\|\boldsymbol{x}\|_q = \left( \sum_{i=1}^N |x_i|^q \right)^{1/q}$ for $1 \leq q < \infty$, and the $L_\infty$ norm is $\|\boldsymbol{x}\|_\infty = \max_{i \in [N]} |x_i|$. The Dirac distribution at $\boldsymbol{x} \in \mathbb{R}^N$ is denoted by $\delta_{\boldsymbol{x}}$. For optimization problems with multiple constraints, we indicate the decision variables on the last line of the constraints.

## 2 Related Work

Research on Markov decision processes under model ambiguity can be traced back to the seventies (Satia and Lave Jr, 1973). In recent years, much progress has been made in the development of

robust MDPs. The most fundamental solution scheme to solve robust MDPs is *robust value iteration*, which is the robust counterpart of the standard value iteration and is first developed in (Givan et al., 2000; Iyengar, 2005; Nilim and El Ghaoui, 2005) for robust MDPs with $(s, a)$-rectangular ambiguity sets. Wiesemann et al. (2013) introduce the $s$-rectangular ambiguity sets; the authors reformulate the robust Bellman updates as convex optimization problems and solve them by using off-the-shelf solvers that have polynomial-time complexity. Researchers have been investigating the theoretical properties of robust MDPs, such as the error bound of applying robust MDPs with state aggregation (Petrik and Subramanian, 2014), the relationship between robustness and regularization (Derman et al., 2021), and the geometry of value function in robust MDPs (Wang et al., 2022). Other than the $s$- and $(s, a)$-ambiguity sets, different types of ambiguity sets have been also proposed to mitigate the side effect of conservatism, such as $k$-rectangularity (Mannor et al., 2016) and $r$-rectangularity (Goyal and Grand-Clément, 2023). Different related models are also proposed to incorporate robustness in different settings, such as robust baseline regret (Ghavamzadeh et al., 2016), imitation learning (Brown et al., 2020), and soft-robust model (Lobo et al., 2020). In terms of algorithmic development, Ho et al. (2018) propose fast algorithms for solving $s$-rectangular robust MDPs with weighted $L_1$ norm ambiguity sets and propose to compute robust Bellman updates using the combination of bisection method and homotopy method, and Behzadian et al. (2021) further extend the solution scheme for the unweighted $L_\infty$ norm cases (Delgado et al., 2016). A *first order method* is introduced in (Grand-Clément and Kroer, 2021b) to approximate robust Bellman updates for robust MDPs with ellipsoidal and Kullback-Leibler (KL) $s$-rectangular ambiguity sets. Recently, Ho et al. (2022) propose the $\phi$-divergence ambiguity set, which generalizes multiple popular choices of ambiguity sets, and they propose an unified solution scheme to compute robust Bellman updates efficiently. On the other hand, *robust policy iteration* alternates between robust policy evaluation and policy improvement (Iyengar, 2005) and it has better empirical performance compared to robust value iteration. Kaufman and Schaefer (2013) propose a modified policy iteration for $(s, a)$-rectangular robust MDPs. Ho et al. (2021) introduce partial policy iteration for $s$- and $(s, a)$-rectangular robust MDPs with $L_1$ norm ambiguity sets. Kumar et al. (2022) study the connection between robust MDPs and regularized MDPs, and propose robust policy iteration for $s$-rectangular robust MDPs. Other than robust policy iteration, Li et al. (2022) propose a policy gradient method for solving rectangular robust MDPs. In order to solve large-scale problems, Tamar et al. (2014) apply value function approximation and propose *robust approximate dynamic programming* for robust MDPs.

While this paper is focused on the model-based setting, recently there is an active line of research focusing on *robust reinforcement learning (RL)* algorithms (Roy et al., 2017; Badrinath and Kalathil, 2021; Wang and Zou, 2021, 2022; Panaganti and Kalathil, 2022). Liu et al. (2022) propose a robust Q-learning algorithm for $(s, a)$-rectangular robust MDPs. A function approximation approach is proposed in (Panaganti et al., 2022) for $(s, a)$-rectangular robust MDPs with offline datasets. The combination of robust RL and deep RL is also studied in (Pinto et al., 2017; Mankowitz et al., 2019; Zhang et al., 2020) although they do not provide theoretical guarantees of the learned policies. We would like to emphasize that both model-based and model-free methods have their own merits: while model-free methods are powerful for applications where data could be generated without much risk and cost, model-based methods are suitable for applications with very limited data or settings where the (time and financial) cost of obtaining new data is high.

For distributionally robust MDPs with the nested-set structured parameter ambiguity sets, Xu and Mannor (2010) propose a Bellman type backward induction to compute the optimal policy, and Yu and Xu (2015) further extend it to a more general state-wise ambiguity set following the unified framework proposed in (Wiesemann et al., 2014). The Wasserstein ambiguity sets are first introduced for distributionally robust MDPs in (Yang, 2017), and a convex formulation is proposed based on Kantorovich duality. Chen et al. (2019) propose a general formulation to combine both moment-based and statistical-distance-based ambiguity sets together for distributionally robust MDPs. Derman and Mannor (2020) study the connection between Wasserstein distributionally robust MDPs and regularization. Abdullah et al. (2019) propose a reinforcement learning framework with distributionally robustiness. As mentioned before, however, (Grand-Clément and Kroer, 2021a) is the only work that focuses on the algorithmic development of solving Wasserstein distributionally robust MDPs, which is also the focus of this paper.

## 3 Preliminaries

A distributionally robust MDP is a tuple $(\mathcal{S}, \mathcal{A}, \boldsymbol{p}_0, \boldsymbol{r}, \lambda, \mathbb{M})$, where $\mathcal{S}$ and $\mathcal{A}$ are the sets of states and actions, respectively. We assume that the state space $\mathcal{S} = \{1, \ldots, S\}$ and the action space $\mathcal{A} = \{1, \ldots, A\}$ are finite. The initial state $s_0$ follows a given initial distribution $\boldsymbol{p}_0 \in \Delta_S$. When the decision maker takes an action $a \in \mathcal{A}$ in state $s \in \mathcal{S}$, the MDP will transit to the next state randomly according to the probability distribution $\boldsymbol{p}_{sa} \in \Delta_S$, which is assumed to be unknown; the transition from state $s \in \mathcal{S}$ under action $a \in \mathcal{A}$ to the next state $s' \in \mathcal{S}$ will induce the reward $r_{sas'}$. We use the vector $\boldsymbol{r}_{sa} \in \mathbb{R}^S$ to represent the connection of all rewards given state $s$ and action $a$. We denote by $\lambda \in (0, 1)$ the discount factor. In distributionally robust MDPs, the transition kernel $\prod_{s \in \mathcal{S}, a \in \mathcal{A}} \boldsymbol{p}_{sa} \in (\Delta_S)^{S \times A}$ is assumed to be uncertain, and it is governed by an unknown probability distribution $\mu$ which is assumed to reside in a prescribed ambiguity set $\mathbb{M}$ that is calibrated from historical data. Note that we also use the notation $\boldsymbol{p}_s = \prod_{a \in \mathcal{A}} \boldsymbol{p}_{sa} \in (\Delta_S)^A$ for the transition kernel given the state $s \in \mathcal{S}$.

The goal of a distributionally robust MDP is to maximize the worst-case expected return; that is,

$$\max_{\boldsymbol{\pi} \in (\Delta_A)^S} \min_{\mu \in \mathbb{M}} \mathbb{E}_{\boldsymbol{\pi}} \left[ \mathbb{E}_{\boldsymbol{p} \sim \mu} \left[ \sum_{t=0}^{\infty} \lambda^t r_{s_t a_t s_{t+1}} \ : \ s_0 \sim \boldsymbol{p}_0 \right] \right]. \tag{1}$$

Here, the decision variable $\boldsymbol{\pi} \in (\Delta_A)^S$ is called a (randomized) policy where for any given $s \in \mathcal{S}$, its sub-vector $\boldsymbol{\pi}_s \in \Delta_A$ represents the probability distribution of taking action $a \in \mathcal{A}$ in state $s$.

It is worth mentioning that the above formulation of distributionally robust MDPs is a generalization of both robust MDPs and standard MDPs. By specifying the ambiguity set $\mathbb{M}$ to be $P(\mathcal{U})$ where $\mathcal{U}$ is the ambiguity set of a robust MDP, the problem (1) is equivalent to the robust MDP with ambiguity set $\mathcal{U}$. If the set $\mathcal{U}$ is a singleton, then problem (1) with $\mathbb{M} = P(\mathcal{U})$ is equivalent to the standard MDP with transition kernel from the singleton $\mathcal{U}$.

Similar to the case of robust MDPs, we consider the common assumption of rectangularity for the sake of tractability, as otherwise problem (1) is NP-hard (Wiesemann et al., 2013). In particular, we consider the following $s$-rectangular $q$-Wasserstein ambiguity set (Grand-Clément and Kroer, 2021a)

$$\mathbb{M}^q = \left\{ \mu \in P\left( (\Delta_S)^{S \times A} \right) \ : \ \mu = \bigotimes_{s \in \mathcal{S}} \mu_s, \ \mu_s \in \mathbb{M}_s^q \subseteq P\left( (\Delta_S)^A \right), \ \forall s \in \mathcal{S} \right\},$$

where $\mathbb{M}_s^q$ is the marginal ambiguity set for the distribution $\mu_s$ that governs the transition kernel at the state $s \in \mathcal{S}$. The superscript $q$ indicates that $\mathbb{M}_s^q$ is defined by $q$-Wasserstein distance; that is,

$$\mathbb{M}_s^q = \left\{ \mu_s \in P\left( (\Delta_S)^A \right) \ : \ W_q\left( \mu_s, \nu_s \right) \leq \theta \right\} \text{ for } 1 \leq q \leq \infty,$$

where $\nu_s$ is the reference distribution at the center of the Wasserstein ball and

$$W_q\left( \mu_s, \nu_s \right) = \min_{\kappa \in P\left( (\Delta_S)^A \times (\Delta_S)^A \right)} \left\{ \left( \mathbb{E}_{(x,y) \sim \kappa} \left( d_q(x, y) \right)^q \right)^{1/q} \ : \ \kappa \in \Gamma(\mu_s, \nu_s) \right\},$$

for $1 \leq q < \infty$, where $\Gamma(\mu_s, \nu_s)$ is the set of all the couplings of $\mu_s$ and $\nu_s$ (Ambrosio and Nicola Gigli, 2005), and $d_q$ is induced by the $q$-norm $d_q(x, y) = \|x - y\|_q$ throughout this work. We define the $\infty$-Wasserstein distance $W_\infty(\mu_s, \nu_s)$ by taking $q \to \infty$.

We adopt the usual setting in distributionally robust data-driven optimization where $\nu_s = \frac{1}{N} \sum_{i=1}^N \delta_{\hat{\boldsymbol{p}}_s^i}$ is set to be the empirical distribution and estimated by the samples $\{\hat{\boldsymbol{p}}_s^i\}_{i=1}^N$ where $\hat{\boldsymbol{p}}_s^i \in (\Delta_S)^A$ for every $i \in [N]$. Then, the optimal value function of (1), $\boldsymbol{v}^\star$, satisfies the following distributionally robust Bellman equation (Yu and Xu, 2015; Yang, 2017; Grand-Clément and Kroer, 2021a)

$$v_s^\star = \max_{\boldsymbol{\pi}_s \in \Delta_A} \min_{\boldsymbol{p}_s \in \mathbb{B}_s^q} \left( \sum_{a \in \mathcal{A}} \pi_{sa} \boldsymbol{p}_{sa}^\top \left( \boldsymbol{r}_{sa} + \lambda \boldsymbol{v}^\star \right) \right) \quad \forall s \in \mathcal{S}, \tag{2}$$

where $\mathbb{B}_s^q$ is the set of expected kernels (Yang, 2017; Bertsimas et al., 2018; Xie, 2020) of the following forms

$$
\begin{aligned}
\mathbb{B}_s^q &= \left\{ \frac{1}{N} \sum_{i=1}^N \boldsymbol{p}_s^i \ : \ \frac{1}{N} \sum_{i=1}^N \left\| \boldsymbol{p}_s^i - \hat{\boldsymbol{p}}_s^i \right\|_q^q \leq \theta^q, \ \boldsymbol{p}_s^i \in (\Delta_S)^A, \ \forall i \in [N] \right\} \quad \text{for } 1 \leq q < \infty, \\
\mathbb{B}_s^\infty &= \left\{ \frac{1}{N} \sum_{i=1}^N \boldsymbol{p}_s^i \ : \ \left\| \boldsymbol{p}_s^i - \hat{\boldsymbol{p}}_s^i \right\|_\infty \leq \theta, \ \boldsymbol{p}_s^i \in (\Delta_S)^A, \ \forall i \in [N] \right\}.
\end{aligned}
\tag{3}
$$

## 4 The Decomposition Algorithm for Distributionally Robust Bellman Update

In this section, we focus on computing the distributionally robust Bellman update, which is the most fundamental operator for solving the distributionally robust Bellman equation (2); we define the distributionally robust Bellman operator $\mathfrak{T}(\cdot)$ where for any $\boldsymbol{v} \in \mathbb{R}^S$,

$$
[\mathfrak{T}(\boldsymbol{v})]_s = \max_{\boldsymbol{\pi}_s \in \Delta_A} \min_{\boldsymbol{p}_s \in \mathbb{B}_s^q} \sum_{a \in \mathcal{A}} \pi_{sa} \cdot \boldsymbol{p}_{sa}^\top (\boldsymbol{r}_{sa} + \lambda \boldsymbol{v}) \quad \forall s \in \mathcal{S}.
\tag{4}
$$

Given the above definition of $\mathfrak{T}(\cdot)$, it is well-known that (2) can be solved by iteratively applying $\mathfrak{T}$, which is a variant of standard value iteration for solving distributionally robust MDPs (Yu and Xu, 2015; Grand-Clément and Kroer, 2021a). For any initial guess $\boldsymbol{v}^0 \in \mathbb{R}^S$, we have $\lim_{t \to \infty} \boldsymbol{v}^t \to \boldsymbol{v}^\star$ where $\boldsymbol{v}^t = \mathfrak{T}(\boldsymbol{v}^{t-1})$ for $t = 1, 2, \dots$ and $\boldsymbol{v}^\star$ satisfies (2). Therefore, the efficiency of evaluating $[\mathfrak{T}(\boldsymbol{v})]_s$ is crucial to the computation of solving distributionally robust MDPs.

However, computing $[\mathfrak{T}(\boldsymbol{v})]_s$ using generic convex optimization solvers is much more computationally demanding compared to the case of classical MDPs, which only have time complexity $\mathcal{O}(SA)$ for computing their Bellman updates. In this section, we exploit the specific problem structure in (4) and reformulate the optimization problem to a form that could be decomposed into smaller problems. We first focus on the case of $q \in [1, \infty)$ in Section 4.1 and then consider the case of $q = \infty$ in Section 4.2, where $q$ indicates the type of Wasserstein distance used in the ambiguity set. As we will show later, combined with the customized fast algorithms in Section 5 for solving the subproblems, one can compute $[\mathfrak{T}(\boldsymbol{v})]_s$ in time complexity that is quasi-linear in $S$ and linear in $A$ and $N$. All the proofs of propositions and theorems in this section are provided in Appendix A.1.

### 4.1 Nested bisection method for $q$-Wasserstein ambiguity set with $q \in [1, \infty)$

We consider the optimization problem in (4) for the case where $q \in [1, \infty)$. By applying the minimax theorem, we obtain the following result.

**Proposition 4.1.** *Consider the Bellman updates* (4) *with* $q \in [1, \infty)$. *Then,*

$$
[\mathfrak{T}(\boldsymbol{v})]_s = \begin{bmatrix} \begin{aligned} \text{minimize} \quad & \gamma \\ \text{subject to} \quad & \frac{1}{N} \sum_{i=1}^N (\boldsymbol{r}_{sa} + \lambda \boldsymbol{v})^\top \boldsymbol{p}_{sa}^i \leq \gamma, \ \forall a \in \mathcal{A} \\ & \frac{1}{N} \sum_{i=1}^N \sum_{a \in \mathcal{A}} \left\| \boldsymbol{p}_{sa}^i - \hat{\boldsymbol{p}}_{sa}^i \right\|_q^q \leq \theta^q \\ & \gamma \in \mathbb{R}, \ \boldsymbol{p}_{sa}^i \in \Delta_S, \ \forall i \in [N], \ \forall a \in \mathcal{A}. \end{aligned} \end{bmatrix} \quad \forall s \in \mathcal{S}.
\tag{5}
$$

The above proposition reformulates the maximin optimization problem in (4) into a convex minimization problem, which can be solved by commercial convex optimization solvers. Moreover, problem (5) can be solved via bisection on $\gamma$; that is, we seek for the lowest possible $\gamma$ such that $\frac{1}{N} \sum_{i=1}^N (\boldsymbol{r}_{sa} + \lambda \boldsymbol{v})^\top \boldsymbol{p}_{sa}^i \leq \gamma$ for each $a \in \mathcal{A}$ while $\boldsymbol{p}_{sa}^i$ satisfies the other constraints in (5), for every $a \in \mathcal{A}$. To this end, we introduce the following subproblem

$$
\mathfrak{P}\left(\{\hat{\boldsymbol{p}}_{sa}^i\}_{i=1}^N; \boldsymbol{b}_{sa}, \gamma\right) = \begin{bmatrix} \begin{aligned} \text{minimize} \quad & \frac{1}{N} \sum_{i=1}^N \left\| \boldsymbol{p}_{sa}^i - \hat{\boldsymbol{p}}_{sa}^i \right\|_q^q \\ \text{subject to} \quad & \frac{1}{N} \sum_{i=1}^N \boldsymbol{b}_{sa}^\top \boldsymbol{p}_{sa}^i \leq \gamma \\ & \boldsymbol{p}_{sa}^i \in \Delta_S, \ \forall i \in [N] \end{aligned} \end{bmatrix}.
\tag{6}
$$

By setting $\boldsymbol{b}_{sa} = \boldsymbol{r}_{sa} + \lambda \boldsymbol{v}$, the above problem $\mathfrak{P}\left(\{\hat{\boldsymbol{p}}_{sa}^i\}_{i=1}^N; \boldsymbol{b}_{sa}, \gamma\right)$ allows us to seek for decision variable $\{\boldsymbol{p}_{sa}^i\}_{i=1}^N$ that satisfies the first and the last line of constraints in (5) while minimizing the "budget" used in the second line of constraint in (5). In particular, for any fixed $\gamma' \in \mathbb{R}$, we distinguish among the following two cases:

(a) If $\sum_{a\in\mathcal{A}} \mathfrak{P}\left(\{\hat{\boldsymbol{p}}_{sa}^i\}_{i=1}^N; \boldsymbol{b}_{sa}, \gamma'\right) \leq \theta^q$, then $\gamma'$ is feasible for (5) and it is an upper bound of the optimal objective value in (5).

(b) If $\sum_{a\in\mathcal{A}} \mathfrak{P}\left(\{\hat{\boldsymbol{p}}_{sa}^i\}_{i=1}^N; \boldsymbol{b}_{sa}, \gamma'\right) > \theta^q$, then $\gamma'$ must be infeasible for problem (5) and so it is a lower bound of the optimal objective value in (5).

To apply the bisection method, we derive the initial upper and lower bounds of the optimal $\gamma^\star$ in (5).

**Proposition 4.2.** *Consider the distributionally robust Bellman update* (5). *The optimal objective value $\gamma^\star$ is bounded by*

$$\max_{a\in\mathcal{A}} \{\min\{\boldsymbol{b}_{sa}\}\} \leq \gamma^\star \leq \max_{a\in\mathcal{A}} \left\{ \frac{1}{N} \sum_{i=1}^N \boldsymbol{b}_{sa}^\top \hat{\boldsymbol{p}}_{sa}^i \right\}. \tag{7}$$

As opposed to (5), the size of problem (6) is independent of $A$. By using the aforementioned bisection method, the overall time complexity is now only linear in $A$ since we only need to solve (6) $\mathcal{O}(A \log \epsilon_1)$ times, where $\epsilon_1$ is the tolerance for the bisection method.

While one can solve problem (6) using off-the-shelf solvers and enjoy the reduced complexity by using the above bisection method, this subproblem itself turns out to be another structural optimization problem. As we will show in the following proposition, by applying duality on (6), the reformulation could be further decomposed via another bisection method on the dual variable.

**Proposition 4.3.** *Consider the problem $\mathfrak{P}\left(\{\hat{\boldsymbol{p}}_{sa}^i\}_{i=1}^N; \boldsymbol{b}_{sa}, \gamma\right)$ in (6). If $\gamma > \min\{\boldsymbol{b}_{sa}\}$, then*

$$\mathfrak{P}\left(\{\hat{\boldsymbol{p}}_{sa}^i\}_{i=1}^N; \boldsymbol{b}_{sa}, \gamma\right) = \max_{0\leq\alpha\leq\bar{\alpha}} -\alpha\gamma + \frac{1}{N} \sum_{i=1}^N \mathfrak{D}_q(\hat{\boldsymbol{p}}_{sa}^i, \boldsymbol{b}_{sa}, \alpha), \tag{8}$$

*where $\bar{\alpha} = \max_{i\in[N]} \left\| \boldsymbol{e}_j - \hat{\boldsymbol{p}}_{sa}^i \right\|_q^q / (\gamma - \min\{\boldsymbol{b}_{sa}\})$ and*

$$\mathfrak{D}_q(\hat{\boldsymbol{p}}_{sa}^i, \boldsymbol{b}_{sa}, \alpha) = \min_{\boldsymbol{p}_{sa}^i \in \Delta_S} \left\| \boldsymbol{p}_{sa}^i - \hat{\boldsymbol{p}}_{sa}^i \right\|_q^q + \alpha \cdot \boldsymbol{b}_{sa}^\top \boldsymbol{p}_{sa}^i, \tag{9}$$

*where $j \in \arg\min_{s'\in\mathcal{S}} b_{sas'}$.*

Notice that we focus on the case where $\gamma > \min\{\boldsymbol{b}_{sa}\}$ since $\min\{\boldsymbol{b}_{sa}\}$ is not larger than the lower bound of $\gamma$ in (7), and the case $\gamma = \min\{\boldsymbol{b}_{sa}\}$ will not occur because $\gamma$ is always taken to be the average of the upper and lower bounds in our bisection method. We refer interested readers to Appendix for more details.

Proposition 4.3 indicates that for any fixed feasible $\alpha$ in (8), the subproblems (9) can be solved separately. Therefore, by applying the bisection method on $\alpha$ in problem (8), one can naturally decompose the problem (8) into $N$ smaller problems that have $S$ variables and $S + 1$ constraints. As a consequence, the overall time complexity is linear in the number of kernels $N$.

By combining both decomposition strategies above, we obtain the proposed nested bisection method, whose pseudocode could be found in Algorithm 1. More details of the algorithm is provided in the Appendix A.1.

**Theorem 4.4.** *Suppose $\gamma'$ is the value returned by Algorithm 1, and $\gamma^\star$ be the optimal value of* (5). *With the inputs provided in Algorithm 1 and user-specified tolerances $\epsilon_1, \epsilon_2 > 0$, we have*

$$|\gamma' - \gamma^\star| \leq \frac{\epsilon_1}{2} + \frac{A\epsilon_2 \left( \max_{a\in\mathcal{A}} \max\{\boldsymbol{b}_{sa}\} - \underline{\gamma} \right) \left( \max_{a\in\mathcal{A}} \max\{\boldsymbol{b}_{sa}\} + \bar{\gamma} \right)}{\theta^q}.$$

**Theorem 4.5.** *Algorithm 1 computes* (5) *in time $\mathcal{O}\left(h_q(S)NA \log \epsilon_1^{-1} \log \epsilon_2^{-1} + AS\right)$, where $h_q(S)$ is the time complexity for solving* (9).

**Algorithm 1:** Nested bisection method to compute distributionally robust Bellman update (5)

---

**Input:** Tolerance $\epsilon_1$ for outer bisection method and $\epsilon_2$ for inner bisection method ;
**Initialization:** Set lower bound $\underline{\gamma}$ and upper bound $\bar{\gamma}$ that are specified in (7) ;
**while** $\bar{\gamma} - \underline{\gamma} > \epsilon_1$ **do**

    Compute $\hat{\gamma} = (\bar{\gamma} + \underline{\gamma})/2$ and $r_0 = 0$ ;
    **for** $a = 1, \ldots, A$ **do**

        Set lower bound $\underline{\alpha}_a = 0$ and compute the upper bound $\bar{\alpha}_a$ in Proposition 4.3 ;
        **while** $\bar{\alpha}_a - \underline{\alpha}_a > \epsilon_2$ **do**

            Compute $\hat{\alpha}_a = (\bar{\alpha}_a + \underline{\alpha}_a)/2$ ;
            Solve $\mathfrak{D}_q(\hat{\boldsymbol{p}}_{sa}^i, \boldsymbol{b}_{sa}, \hat{\alpha}_a)$, for every $i \in [N]$ ;
            Compute the slope of (8): $m = -\hat{\gamma} + \frac{1}{N}\sum_{i=1}^N \boldsymbol{b}_{sa}^\top \boldsymbol{p}_{sa}^{i,\star}$ ;
            **if** $m \leq 0$ **then** set $\bar{\alpha}_a = \hat{\alpha}_a$ **else** set $\underline{\alpha}_a = \hat{\alpha}_a$;
        **end**
        Set $\alpha_a^\star = (\bar{\alpha}_a + \underline{\alpha}_a)/2$ ;
        Solve $\mathfrak{D}_q(\hat{\boldsymbol{p}}_{sa}^i, \boldsymbol{b}_{sa}, \alpha_a^\star)$, for every $i \in [N]$ ;
        Set $r_a = r_{a-1} - \alpha_a^\star \hat{\gamma} + \frac{1}{N}\sum_{i=1}^N \mathfrak{D}_q(\hat{\boldsymbol{p}}_{sa}^i, \boldsymbol{b}_{sa}, \alpha_a^\star)$;
    **end**
    **if** $r_A > \theta^q$ **then** set $\underline{\gamma} = \hat{\gamma}$ **else** set $\bar{\gamma} = \hat{\gamma}$;
**end**
**Result:** Optimal objective value of (5): $\gamma = (\bar{\gamma} + \underline{\gamma})/2$;

---

As shown in the above theorem, the proposed nested bisection method has a time complexity that is linear in both $A$ and $N$, but the overall complexity depends on the subproblem (9) where its complexity depends on the choice of $q$ and the number of states $S$. In Section 5, we will derive $h_1(S)$ and $h_2(S)$, which are the two most common cases. The complexities associated with solving equations (5) and (8) using general convex optimization methods are also provided in the Appendix B.

### 4.2 Decomposition scheme for $\infty$-Wasserstein ambiguity set

As opposed to the case where $q \in [1, \infty)$, for $\infty$-Wasserstein ambiguity set, the corresponding Bellman update can be naturally decomposed without using any bisection method.

**Proposition 4.6.** *Consider the Bellman update* (4) *with* $q = \infty$. *Then,*

$$[\mathfrak{T}(\boldsymbol{v})]_s = \frac{1}{N}\max_{a\in\mathcal{A}}\sum_{i=1}^N \min_{\boldsymbol{p}_{sa}^i\in\Delta_S}\left\{\boldsymbol{b}_{sa}^\top\boldsymbol{p}_{sa}^i \ : \ \left\|\boldsymbol{p}_{sa}^i - \hat{\boldsymbol{p}}_{sa}^i\right\|_\infty \leq \theta\right\} \quad \forall s \in \mathcal{S}. \tag{10}$$

The above reformulation requires solving $NA$ inner minimization problems that have $S$ variables and $S + 2$ constraints. Hence, the time complexity of Bellman update (10) is linear in $A$ and $N$. Similar to Section 4.1, we will discuss the complexity of solving the inner minimization in the next section.

## 5 Efficient Algorithms for Subproblems

Section 4 offers decomposition schemes for distributionally robust MDPs with $q$-Wasserstein ambiguity set. As shown in Theorem 4.5 and Proposition 4.6, the overall complexity of computing distributionally robust Bellman update is linear in both $A$ and $N$ but it depends on complexities of solving the subproblem (9) and the inner minimization problem in (10). In this section, we consider the common cases where $q \in \{1, 2, \infty\}$ and discuss the time complexity of solving the subproblem in each case.

As we will show below, the overall time complexities of our decomposition algorithms are only $\mathcal{O}\left(SA\log S + NAS\log\epsilon_1^{-1}\log\epsilon_2^{-1}\right)$ for $q = 1$, $\mathcal{O}(NAS\log S\log\epsilon_1^{-1}\log\epsilon_2^{-1})$ for $q = 2$, and $\mathcal{O}(AS(\log S + N))$ when $q = \infty$. These complexities are much lower than $\mathcal{O}(N^{3.5}A^{3.5}S^{3.5}\log(\epsilon^{-1}))$ and $\mathcal{O}(NA^{2.5}S^{2.5}\log(S)\epsilon^{-1.5})$ from the state-of-the-art solution methods in (Xu and Mannor, 2010) and (Grand-Clément and Kroer, 2021a), respectively. The proofs of

the following propositions, theorems, and corollaries and the details of the proposed algorithms are relegated to Appendix A.2.

## 5.1    1-Wasserstein ambiguity set

When $q = 1$, the corresponding subproblem (9) has an equivalent form as follows.

**Proposition 5.1.** *Suppose $q = 1$. The minimization problem* (9) *is equivalent to*

$$\min_{v \geq 0} \left\{ v + \alpha \min_{\boldsymbol{p}_{sa}^i \in \Delta_S} \boldsymbol{b}_{sa}^\top \boldsymbol{p}_{sa}^i \ : \ \|\boldsymbol{p}_{sa}^i - \hat{\boldsymbol{p}}_{sa}^i\|_1 \leq v \right\}. \tag{11}$$

While problem (11) does not appear to be trivial at first glance, it has nice mathematical properties which allow us to solve this problem efficiently.

**Theorem 5.2.** *The objective function in* (11) *is piecewise-linear in $v$, and it can be solved in time $\mathcal{O}(S \log S)$.*

Therefore, the proposed Algorithm 1 can compute the Bellman update with the time complexity linear in $A$ and $N$ and quali-linear in $S$. We obtain the following result by combining Theorem 4.5 and Theorem 5.2.

**Corollary 5.2.1.** *The distributionally robust Bellman update with $1$-Wasserstein ambiguity set can be computed in time $\mathcal{O}\left(SA \log S + NAS \log \epsilon_1^{-1} \log \epsilon_2^{-1}\right)$, where $\epsilon_1$ and $\epsilon_2$ are the user-specified tolerances in Algorithm 1.*

## 5.2    2-Wasserstein ambiguity set

When $q = 2$, the following result shows that (9) could be transformed to an Euclidean projection problem onto the probability simplex.

**Theorem 5.3.** *Suppose $q = 2$. The minimization problem* (9) *is equivalent to*

$$-\frac{\alpha^2 \|\boldsymbol{b}_{sa}\|_2^2}{4} + \alpha \boldsymbol{b}_{sa}^\top \hat{\boldsymbol{p}}_{sa}^i + \min_{\boldsymbol{p}_{sa}^i \in \Delta_S} \left\| \boldsymbol{p}_{sa}^i - \frac{2\hat{\boldsymbol{p}}_{sa}^i - \alpha \boldsymbol{b}_{sa}}{2} \right\|_2^2, \tag{12}$$

*which can be solved in time $\mathcal{O}(S \log S)$.*

Euclidean projection problem onto the probability simplex can be solved via the existing algorithm (Wang and Carreira-Perpinán, 2013). By combining the results in Theorem 4.5 and Theorem 5.3, we obtain the following result.

**Corollary 5.3.1.** *The distributionally robust Bellman update with $2$-Wasserstein ambiguity set can be computed in time $\mathcal{O}\left(NAS \log S \log \epsilon_1^{-1} \log \epsilon_2^{-1}\right)$, where $\epsilon_1$ and $\epsilon_2$ are the user-specified tolerances in Algorithm 1.*

## 5.3    $\infty$-Wasserstein ambiguity set

When $q = \infty$, the corresponding subproblem

$$\min_{\boldsymbol{p}_{sa}^i \in \Delta_S} \left\{ \boldsymbol{b}_{sa}^\top \boldsymbol{p}_{sa}^i \ : \ \left\| \boldsymbol{p}_{sa}^i - \hat{\boldsymbol{p}}_{sa}^i \right\|_\infty \leq \theta \right\} \tag{13}$$

can be reformulated to a linear program with box constraints and a linear equality constraint which can be solved efficiently.

**Theorem 5.4.** *The minimization problem* (13) *is solved in time $\mathcal{O}(S \log S)$.*

Therefore, we obtain the overall complexity as follows.

**Corollary 5.4.1.** *The distributionally robust Bellman update with $\infty$-Wasserstein ambiguity set can be computed in time $\mathcal{O}\left(AS \left(\log S + N\right)\right)$.*

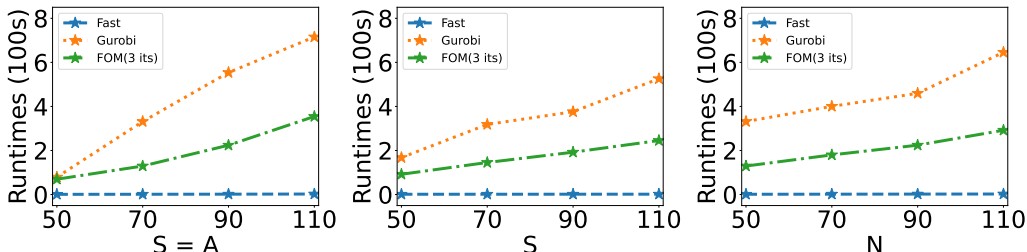

Figure 1: Comparisons of all algorithms for $q = 1$, where *(left)* $N = 50$, *(middle)* $A = 70$, $N = 50$, and *(right)* $S = A = 70$.

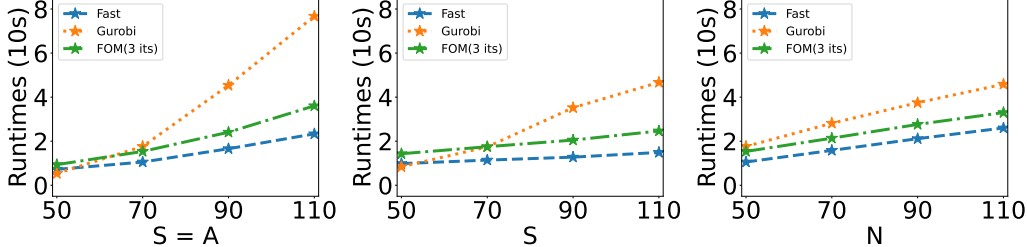

Figure 2: Comparisons of all algorithms for $q = 2$, where *(left)* $N = 50$, *(middle)* $A = 70$, $N = 50$, and *(right)* $S = A = 70$.

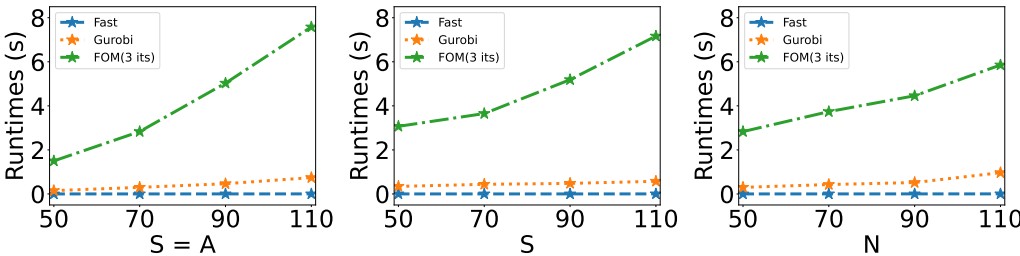

Figure 3: Comparisons of all algorithms for $q = \infty$, where *(left)* $N = 50$, *(middle)* $A = 70$, $N = 50$, and *(right)* $S = A = 70$.

# 6 Numerical Results

We experiment on the performance of the proposed algorithms (Fast), Gurobi (Gurobi Optimization, LLC, 2023), and the first-order method (FOM) proposed in (Grand-Clément and Kroer, 2021a) with different sizes of the randomly generated distributionally robust MDPs. Due to page limit, we only report the results of the experiments on a single Bellman update, and we provide additional experimental results in the Appendix C, which also contains the detailed settings of all the experiments.

Figure 1 reports the average computation times of Bellman updates with 1-Wasserstein ambiguity sets. For the first-order method (Grand-Clément and Kroer, 2021a), we report the computation times of the third Bellman iteration, and these computation times increase at every Bellman iteration of the first-order method. One can see that all three algorithms perform similarly when problem size is small. However, as $S$, $A$, or $N$ increases, the runtimes of both Gurobi and first-order method increase rapidly, while the proposed algorithm remains scalable.

The results with 2-Wasserstein ambiguity sets and $\infty$-Wasserstein ambiguity sets are shown in Figure 2 and Figure 3, respectively. These results are similar to the case where $q = 1$, which are consistent to our theoretical results on complexities. As expected, the proposed algorithms are several orders of magnitude faster than the existing state-of-the-art solution methods.

# 7    Conclusion

This paper studies distributionally robust MDPs with Wasserstein ambiguity sets. In particular, we propose fast algorithms to compute distributionally robust Bellman updates. We show that the proposed algorithms outperform other existing methods in both theory and experiments. Future work could address the extensions to approximate dynamic programming for distributionally robust MDPs as well as model-free settings.

# 8    Acknowledgement

We thank the anonymous reviewers for their supportive comments. This work was supported, in part, by the CityU Start-Up Grant (Project No. 9610481), the National Natural Science Foundation of China (Project No. 72032005), and Chow Sang Sang Group Research Fund sponsored by Chow Sang Sang Holdings International Limited (Project No. 9229076).

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

# A Appendix: Proofs and Algorithms

## A.1 Proofs of results in Section 4

*Proof of Proposition 4.1.* Plug $\mathbb{B}_s^q$ in (3) into (4), and apply the minimax theorem, the original problem $[\mathfrak{T}(\boldsymbol{v})]_s$, $\forall s \in \mathcal{S}$ is given by:

$$
\max_{\boldsymbol{\pi}_s \in \Delta_A} \quad \min_{\boldsymbol{p}_s^1, \cdots, \boldsymbol{p}_s^N} \sum_{a \in \mathcal{A}} \pi_{sa} \left( \frac{1}{N} \sum_{i=1}^N \boldsymbol{p}_{sa}^i \right)^\top (\boldsymbol{r}_{sa} + \lambda \boldsymbol{v})
$$

$$
\text{s.t.} \quad \frac{1}{N} \sum_{i=1}^N \left\| \boldsymbol{p}_s^i - \hat{\boldsymbol{p}}_s^i \right\|_q^q \leq \theta^q,
$$

$$
\boldsymbol{p}_{sa}^i \in \Delta_S, \ \forall i \in [N], \ \forall a \in \mathcal{A}
$$

$$
= \quad \min \quad \max_{\boldsymbol{\pi}_s \in \Delta_A} \frac{1}{N} \sum_{a \in \mathcal{A}} \pi_{sa} \sum_{i=1}^N (\boldsymbol{r}_{sa} + \lambda \boldsymbol{v})^\top \boldsymbol{p}_{sa}^i
$$

$$
\text{s.t.} \quad \frac{1}{N} \sum_{i=1}^N \left\| \boldsymbol{p}_s^i - \hat{\boldsymbol{p}}_s^i \right\|_q^q \leq \theta^q,
$$

$$
\boldsymbol{p}_{sa}^i \in \Delta_S, \ \forall i \in [N], \ \forall a \in \mathcal{A}
$$

$$
= \quad \min \quad \max_{a \in \mathcal{A}} \frac{1}{N} \sum_{i=1}^N (\boldsymbol{r}_{sa} + \lambda \boldsymbol{v})^\top \boldsymbol{p}_{sa}^i
$$

$$
\text{s.t.} \quad \frac{1}{N} \sum_{i=1}^N \sum_{a \in \mathcal{A}} \left\| \boldsymbol{p}_s^i - \hat{\boldsymbol{p}}_s^i \right\|_q^q \leq \theta^q,
$$

$$
\boldsymbol{p}_{sa}^i \in \Delta_S, \ \forall i \in [N], \ \forall a \in \mathcal{A}.
$$

Hence (5) is a direct consequence of above formulation by introducing the following epigraph variable $\gamma$ which satisfies $\gamma \geq \max_{a \in \mathcal{A}} \frac{1}{N} \sum_{i=1}^N (\boldsymbol{r}_{sa} + \lambda \boldsymbol{v})^\top \boldsymbol{p}_{sa}^i$. $\qquad \square$

*Proof of Proposition 4.2.* To prove the upper bound of $\gamma^\star$, we consider $\bar{\gamma} = \max_{a \in \mathcal{A}} \frac{1}{N} \sum_{i=1}^N \boldsymbol{b}_{sa}^\top \hat{\boldsymbol{p}}_{sa}^i$, which equals to $\bar{\gamma} \geq \frac{1}{N} \sum_{i=1}^N \boldsymbol{b}_{sa}^\top \hat{\boldsymbol{p}}_{sa}^i$, $\forall a \in \mathcal{A}$. This implies $\{\hat{\boldsymbol{p}}_{sa}^i\}_{i=1}^N$ satisfies every constraint in problem (6) with the lowest possible objective value 0, for every $a \in \mathcal{A}$. Therefore, $\sum_{a \in \mathcal{A}} \mathfrak{P}\left(\{\hat{\boldsymbol{p}}_{sa}^i\}_{i=1}^N; \boldsymbol{r}_{sa} + \lambda \boldsymbol{v}, \bar{\gamma}\right) = 0 < \theta^q$. At this time, $\bar{\gamma}$ is feasible for (5). Hence, we provide an upper bound $\bar{\gamma} = \max_{a \in \mathcal{A}} \frac{1}{N} \sum_{i=1}^N \boldsymbol{b}_{sa}^\top \hat{\boldsymbol{p}}_{sa}^i$.

To prove the lower bound of $\gamma^\star$, we assume the contrary $\max_{a \in \mathcal{A}} \{\min \{\boldsymbol{b}_{sa}\}\} > \gamma^\star$. So there exists $\hat{a} \in \mathcal{A}$ with $\gamma^\star < \min \{\boldsymbol{b}_{s\hat{a}}\}$. Then there is no $\boldsymbol{p}_{s\hat{a}}^i$ satisfies the first constraint in (5) for $a = \hat{a}$. This contradiction verifies the lower bound $\underline{\gamma} = \max_{a \in \mathcal{A}} \{\min \{\boldsymbol{b}_{sa}\}\}$.

$\qquad \square$

*Proof of Proposition 4.3.* By definition, for any fixed $s \in \mathcal{S}$ and $a \in \mathcal{A}$, we have

$$
\mathfrak{P}\left(\{\hat{\boldsymbol{p}}_{sa}^i\}_{i=1}^N; \boldsymbol{b}_{sa}, \gamma\right) = \min \quad \frac{1}{N} \sum_{i=1}^N \left\| \boldsymbol{p}_{sa}^i - \hat{\boldsymbol{p}}_{sa}^i \right\|_q^q
$$

$$
\text{s.t.} \quad \frac{1}{N} \sum_{i=1}^N \boldsymbol{b}_{sa}^\top \boldsymbol{p}_{sa}^i \leq \gamma,
$$

$$
\boldsymbol{p}_{sa}^i \in \Delta_S, \ \forall i \in [N].
$$

Then, we introduce the dual variable $\alpha \geq 0$ for the constraint $\frac{1}{N} \sum_{i=1}^N \boldsymbol{b}_{sa}^\top \boldsymbol{p}_{sa}^i \leq \gamma$ and obtain

$$
\mathfrak{P}\left(\{\hat{\boldsymbol{p}}_{sa}^i\}_{i=1}^N; \boldsymbol{b}_{sa}, \gamma\right) = \max_{\alpha \geq 0} \left\{ \min_{\boldsymbol{p}_{sa}^i \in \Delta_S, \forall i \in [N]} \frac{1}{N} \sum_{i=1}^N \left\| \boldsymbol{p}_{sa}^i - \hat{\boldsymbol{p}}_{sa}^i \right\|_q^q + \alpha \left( \frac{1}{N} \sum_{i=1}^N \boldsymbol{b}_{sa}^\top \boldsymbol{p}_{sa}^i - \gamma \right) \right\}
$$

$$
= \max_{\alpha \geq 0} \quad -\alpha \gamma + \frac{1}{N} \sum_{i=1}^N \mathfrak{D}_q(\hat{\boldsymbol{p}}_{sa}^i, \boldsymbol{b}_{sa}, \alpha),
$$

where $\mathfrak{D}_q(\hat{\boldsymbol{p}}_{sa}^i, \boldsymbol{b}_{sa}, \alpha)$ is defined in the Proposition 4.3.

To show $\bar{\alpha}$ defined in the proposition is indeed an upper bound of the optimal $\alpha^\star$, we denote

$$f(\alpha) = -\alpha\gamma + \frac{1}{N}\sum_{i=1}^{N}\mathfrak{D}_q(\hat{\boldsymbol{p}}_{sa}^i, \boldsymbol{b}_{sa}, \alpha).$$

Notice that $\forall \alpha \geq \max_{i\in[N]} \frac{\|\boldsymbol{e}_j - \hat{\boldsymbol{p}}_{sa}^i\|_q^q}{\gamma - \min\{\boldsymbol{b}_{sa}\}}$, where $j \in \underset{s'\in\mathcal{S}}{\arg\min}\, b_{sas'}$ :

$$f(\alpha) \leq -\alpha\gamma + \frac{1}{N}\sum_{i=1}^{N}\left\|\boldsymbol{e}_j - \hat{\boldsymbol{p}}_{sa}^i\right\|_q^q + \alpha\boldsymbol{b}_{sa}^\top\boldsymbol{e}_j$$

$$= \frac{1}{N}\sum_{i=1}^{N}\left\|\boldsymbol{e}_j - \hat{\boldsymbol{p}}_{sa}^i\right\|_q^q + (\min\{\boldsymbol{b}_{sa}\} - \gamma)\,\alpha$$

$$\leq 0 = f(0),$$

where the first inequality is from the definition of $\mathfrak{D}_q(\hat{\boldsymbol{p}}_{sa}^i, \boldsymbol{b}_{sa}, \alpha)$, the second equality is due to the selection of $j$, and the last inequality is due to the selection of $\alpha$. Hence $\alpha^\star \in [0, \bar{\alpha}]$ by $f(\alpha)$ is concave w.r.t. $\alpha$ (otherwise $\exists \alpha^\star > \bar{\alpha}$ such that $f(\alpha^\star) > f(0) \geq f(\bar{\alpha})$, where nonconcavity of $f$ follows), and (8) is true. One can further compute the subdifferential of $f(\alpha)$ by Danskin's theorem (Bertsekas, 1999). Typically,

$$-\gamma + \frac{1}{N}\sum_{i=1}^{N}\boldsymbol{b}_{sa}^\top\boldsymbol{p}_{sa}^{i,\star} \in \partial f(\alpha),$$

where $\boldsymbol{p}_{sa}^{i,\star}$ is any minimizer of the inner minimization problem corresponding with given $\alpha \geq 0$. $\quad\square$

*Proof of Theorem 4.4.* For simplicity of notations, throughout this proof we use

$$\delta \triangleq \frac{A\epsilon_2}{2}\left(\max_{a\in\mathcal{A}}\max\{\boldsymbol{b}_{sa}\} + \bar{\gamma}\right) \qquad \text{and} \qquad f(\gamma) \triangleq \sum_{a\in\mathcal{A}}\mathfrak{P}\left(\{\hat{\boldsymbol{p}}_{sa}^i\}_{i=1}^N; \boldsymbol{b}_{sa}, \gamma\right), \quad \forall\gamma \in [\underline{\gamma}, \bar{\gamma}].$$

By (6), we can get that $\mathfrak{P}\left(\{\hat{\boldsymbol{p}}_{sa}^i\}_{i=1}^N; \boldsymbol{b}_{sa}, \gamma\right)$ and $f(\gamma)$ are non-increasing in $[\underline{\gamma}, \bar{\gamma}]$. For each given $\gamma \in [\underline{\gamma}, \bar{\gamma}]$ and $a \in \mathcal{A}$, we denote $\hat{\mathfrak{P}}\left(\{\hat{\boldsymbol{p}}_{sa}^i\}_{i=1}^N; \boldsymbol{b}_{sa}, \gamma\right)$ the corresponding value calculated by the Algorithm 1. Furthermore, we call

$$\hat{f}(\gamma) \triangleq \sum_{a\in\mathcal{A}}\hat{\mathfrak{P}}\left(\{\hat{\boldsymbol{p}}_{sa}^i\}_{i=1}^N; \boldsymbol{b}_{sa}, \gamma\right), \quad \forall\gamma \in [\underline{\gamma}, \bar{\gamma}].$$

We can show that

$$|f(\gamma) - \hat{f}(\gamma)| \leq \delta, \quad \forall\gamma \in [\underline{\gamma}, \bar{\gamma}]. \tag{14}$$

We fix $\gamma \in [\underline{\gamma}, \bar{\gamma}]$, and consider (8). Algorithm 1 provides the optimal solution for (8) with tolerance $\epsilon_2/2$, so we have $|\alpha_{sa,\gamma}^\star - \hat{\alpha}_{sa,\gamma}| \leq \epsilon_2/2$, where $\alpha_{sa,\gamma}^\star$ is the true optimal solution of (8) and $\hat{\alpha}_{sa,\gamma}$ is the solution computed in the inner bisection in Algorithm 1. Thus we get

$$\left|\mathfrak{P}\left(\{\hat{\boldsymbol{p}}_{sa}^i\}_{i=1}^N; \boldsymbol{b}_{sa}, \gamma\right) - \hat{\mathfrak{P}}\left(\{\hat{\boldsymbol{p}}_{sa}^i\}_{i=1}^N; \boldsymbol{b}_{sa}, \gamma\right)\right|$$

$$= -\alpha_{sa,\gamma}^\star\gamma + \frac{1}{N}\sum_{i=1}^{N}\mathfrak{D}_q(\hat{\boldsymbol{p}}_{sa}^i, \boldsymbol{b}_{sa}, \alpha_{sa,\gamma}^\star) - \left(-\hat{\alpha}_{sa,\gamma}\gamma + \frac{1}{N}\sum_{i=1}^{N}\mathfrak{D}_q(\hat{\boldsymbol{p}}_{sa}^i, \boldsymbol{b}_{sa}, \hat{\alpha}_{sa,\gamma})\right)$$

$$\leq \left|\alpha_{sa,\gamma}^\star - \hat{\alpha}_{sa,\gamma}\right|\gamma + \frac{1}{N}\sum_{i=1}^{N}\left|\mathfrak{D}_q(\hat{\boldsymbol{p}}_{sa}^i, \boldsymbol{b}_{sa}, \alpha_{sa,\gamma}^\star) - \mathfrak{D}_q(\hat{\boldsymbol{p}}_{sa}^i, \boldsymbol{b}_{sa}, \hat{\alpha}_{sa,\gamma})\right|$$

$$\leq \frac{\epsilon_2}{2}\left(\bar{\gamma} + \max_{a\in\mathcal{A}}\max\{\boldsymbol{b}_{sa}\}\right),$$

where the last step is due to that

$$\mathfrak{D}_q(\hat{\boldsymbol{p}}_{sa}^i, \boldsymbol{b}_{sa}, \alpha_{sa,\gamma}^\star) - \mathfrak{D}_q(\hat{\boldsymbol{p}}_{sa}^i, \boldsymbol{b}_{sa}, \hat{\alpha}_{sa,\gamma})$$

$$= \min_{\boldsymbol{p}_{sa}^i \in \Delta_S} \left\| \boldsymbol{p}_{sa}^i - \hat{\boldsymbol{p}}_{sa}^i \right\|_q^q + \alpha_{sa,\gamma}^\star \cdot \boldsymbol{b}_{sa}^\top \boldsymbol{p}_{sa}^i - \left( \min_{\boldsymbol{p}_{sa}^i \in \Delta_S} \left\| \boldsymbol{p}_{sa}^i - \hat{\boldsymbol{p}}_{sa}^i \right\|_q^q + \hat{\alpha}_{sa,\gamma} \cdot \boldsymbol{b}_{sa}^\top \boldsymbol{p}_{sa}^i \right)$$

$$\leq \left( \alpha_{sa,\gamma}^\star - \hat{\alpha}_{sa,\gamma} \right) \boldsymbol{b}_{sa}^\top \boldsymbol{p}_{sa}^{i,\hat{\alpha}}$$

$$\leq \frac{\epsilon_2}{2} \max_{a \in \mathcal{A}} \max\{\boldsymbol{b}_{sa}\},$$

and

$$\mathfrak{D}_q(\hat{\boldsymbol{p}}_{sa}^i, \boldsymbol{b}_{sa}, \hat{\alpha}_{sa,\gamma}) - \mathfrak{D}_q(\hat{\boldsymbol{p}}_{sa}^i, \boldsymbol{b}_{sa}, \alpha_{sa,\gamma}^\star)$$

$$= \min_{\boldsymbol{p}_{sa}^i \in \Delta_S} \left\| \boldsymbol{p}_{sa}^i - \hat{\boldsymbol{p}}_{sa}^i \right\|_q^q + \hat{\alpha}_{sa,\gamma} \cdot \boldsymbol{b}_{sa}^\top \boldsymbol{p}_{sa}^i - \left( \min_{\boldsymbol{p}_{sa}^i \in \Delta_S} \left\| \boldsymbol{p}_{sa}^i - \hat{\boldsymbol{p}}_{sa}^i \right\|_q^q + \alpha^\star \cdot \boldsymbol{b}_{sa}^\top \boldsymbol{p}_{sa}^i \right)$$

$$\leq \left( \hat{\alpha}_{sa,\gamma} - \alpha_{sa,\gamma}^\star \right) \boldsymbol{b}_{sa}^\top \boldsymbol{p}_{sa}^{i,\alpha^\star}$$

$$\leq \frac{\epsilon_2}{2} \max_{a \in \mathcal{A}} \max\{\boldsymbol{b}_{sa}\},$$

here $\boldsymbol{p}_{sa}^{i,\hat{\alpha}}$ and $\boldsymbol{p}_{sa}^{i,\alpha^\star}$ are the optimal solutions to (9) for $\alpha = \hat{\alpha}_{sa,\gamma}$ and $\alpha = \alpha_{sa,\gamma}^\star$ respectively.

Hence, (14) is the direct consequence of above estimation, together with the definitions of $f$, $\hat{f}$ and $\delta$. And we have $\gamma^\star = \inf\{\gamma \in [\underline{\gamma}, \bar{\gamma}] : f(\gamma) \leq \theta^q\}$. We further define $\hat{\gamma} \triangleq \inf\{\gamma \in [\underline{\gamma}, \bar{\gamma}] : \hat{f}(\gamma) \leq \theta^q\}$. The outer bisection of Algorithm 1 implies that $|\gamma' - \hat{\gamma}| \leq \frac{\epsilon_1}{2}$, so to prove the claimed result in the theorem, it suffices to show that

$$|\gamma^\star - \hat{\gamma}| \leq \frac{2\delta \left( \max_{a \in \mathcal{A}} \max\{\boldsymbol{b}_{sa}\} - \underline{\gamma} \right)}{\theta^q}. \tag{15}$$

Notice that $\hat{\mathfrak{P}}\left(\{\hat{\boldsymbol{p}}_{sa}^i\}_{i=1}^N; \boldsymbol{b}_{sa}, \gamma\right) \leq \mathfrak{P}\left(\{\hat{\boldsymbol{p}}_{sa}^i\}_{i=1}^N; \boldsymbol{b}_{sa}, \gamma\right), \forall \gamma \in [\underline{\gamma}, \bar{\gamma}]$ by definitions, we get $\hat{f}(\gamma) \leq f(\gamma)$, so $\{\gamma \in [\underline{\gamma}, \bar{\gamma}] : f(\gamma) \leq \theta^q\} \subseteq \{\gamma \in [\underline{\gamma}, \bar{\gamma}] : \hat{f}(\gamma) \leq \theta^q\}$, hence we get $\hat{\gamma} \leq \gamma^\star$.

We assume that $\gamma^\star - \underline{\gamma} \geq \frac{2\delta \left( \max_{a \in \mathcal{A}} \max\{\boldsymbol{b}_{sa}\} - \underline{\gamma} \right)}{\theta^q}$, otherwise (15) is trivially satisfied. To achieve (15), we claim the following statement:

$$f\left( \gamma^\star - \frac{2\delta \left( \max_{a \in \mathcal{A}} \max\{\boldsymbol{b}_{sa}\} - \underline{\gamma} \right)}{\theta^q} \right) > \theta^q + \delta. \tag{16}$$

If the statement is true, we have $\forall \gamma \in [\underline{\gamma}, \gamma^\star - \left( \max_{a \in \mathcal{A}} \max\{\boldsymbol{b}_{sa}\} - \underline{\gamma} \right)(2\delta/\theta^q)]$:

$$\hat{f}(\gamma) \geq f(\gamma) - \delta \geq f\left( \gamma^\star - \frac{2\delta \left( \max_{a \in \mathcal{A}} \max\{\boldsymbol{b}_{sa}\} - \underline{\gamma} \right)}{\theta^q} \right) - \delta > \theta^q,$$

where the first inequality is from (14) and $\hat{f}(\gamma) \leq f(\gamma)$, the second inequality is due to that $f(\gamma)$ is non-increasing, and the third inequality is from the above statement.

So $\{\gamma \in [\underline{\gamma}, \bar{\gamma}] : \hat{f}(\gamma) \leq \theta^q\} \subseteq (\gamma^\star - \left( \max_{a \in \mathcal{A}} \max\{\boldsymbol{b}_{sa}\} - \underline{\gamma} \right)(2\delta/\theta^q), \bar{\gamma}]$, thus

$$\gamma^\star - \frac{2\delta \left( \max_{a \in \mathcal{A}} \max\{\boldsymbol{b}_{sa}\} - \underline{\gamma} \right)}{\theta^q} \leq \hat{\gamma} \leq \gamma^\star,$$

which implies the desired (15).

For the proof of the statement (16). We argue that

$$\forall \gamma \in [\gamma^\star - \frac{2\delta \left( \max_{a \in \mathcal{A}} \max\{\boldsymbol{b}_{sa}\} - \underline{\gamma} \right)}{\theta^q}, \gamma^\star) : \sum_{a \in \mathcal{A}} \alpha_{sa,\gamma}^\star \geq \frac{\theta^q}{\left( \max_{a \in \mathcal{A}} \max\{\boldsymbol{b}_{sa}\} - \underline{\gamma} \right)}, \tag{17}$$

by contradiction. Assume there is some $\gamma'' \in [\gamma^\star - \frac{2\delta\left(\max_{a\in\mathcal{A}}\max\{\boldsymbol{b}_{sa}\}-\underline{\gamma}\right)}{\theta^q}, \gamma^\star)$ with that $\sum_{a\in\mathcal{A}}\alpha^\star_{sa,\gamma''} < \frac{\theta^q}{\left(\max_{a\in\mathcal{A}}\max\{\boldsymbol{b}_{sa}\}-\underline{\gamma}\right)}$. Clearly $f(\gamma'') > \theta^q$ since $\gamma'' < \gamma^\star$. Then

$$\theta^q > \sum_{a\in\mathcal{A}}\alpha^\star_{sa,\gamma''}\left(\max_{a\in\mathcal{A}}\max\{\boldsymbol{b}_{sa}\}-\underline{\gamma}\right)$$

$$\geq \sum_{a\in\mathcal{A}}\alpha^\star_{sa,\gamma''}\left(\frac{1}{N}\sum_{i=1}^{N}\left(\boldsymbol{b}_{sa}^\top\hat{\boldsymbol{p}}^i_{sa}-\gamma\right)\right)$$

$$= \sum_{a\in\mathcal{A}}\frac{1}{N}\sum_{i=1}^{N}\alpha^\star_{sa,\gamma''}\left(\boldsymbol{b}_{sa}^\top\hat{\boldsymbol{p}}^i_{sa}-\gamma\right)$$

$$\geq \sum_{a\in\mathcal{A}}\frac{1}{N}\sum_{i=1}^{N}\min_{\boldsymbol{p}^i_{sa}\in\Delta_S}\left\|\boldsymbol{p}^i_{sa}-\hat{\boldsymbol{p}}^i_{sa}\right\|_q^q+\alpha^\star_{sa,\gamma''}\cdot\left(\boldsymbol{b}_{sa}^\top\boldsymbol{p}^i_{sa}-\gamma\right)$$

$$= \sum_{a\in\mathcal{A}}-\alpha^\star_{sa,\gamma''}\gamma''+\frac{1}{N}\sum_{i=1}^{N}\mathfrak{D}_q(\hat{\boldsymbol{p}}^i_{sa},\boldsymbol{b}_{sa},\alpha^\star_{sa,\gamma''})$$

$$= \sum_{a\in\mathcal{A}}\mathfrak{P}\left(\{\hat{\boldsymbol{p}}^i_{sa}\}_{i=1}^N;\boldsymbol{b}_{sa},\gamma''\right)$$

$$= f(\gamma''),$$

which is contradicted with $f(\gamma'') > \theta^q$. This contradiction implies that (17) is true.
For simplicity of notations, we call $\gamma''' = \gamma^\star - \frac{2\delta\left(\max_{a\in\mathcal{A}}\max\{\boldsymbol{b}_{sa}\}-\underline{\gamma}\right)}{\theta^q}$, and we fix any $\gamma^\ell \in (\gamma''',\gamma^\star)$, then

$$\theta^q - f\left(\gamma^\star - \frac{2\delta\left(\max_{a\in\mathcal{A}}\max\{\boldsymbol{b}_{sa}\}-\underline{\gamma}\right)}{\theta^q}\right)$$

$$< f(\gamma^\ell) - f(\gamma''')$$

$$= \sum_{a\in\mathcal{A}}\left(\mathfrak{P}\left(\{\hat{\boldsymbol{p}}^i_{sa}\}_{i=1}^N;\boldsymbol{b}_{sa},\gamma^\ell\right)-\mathfrak{P}\left(\{\hat{\boldsymbol{p}}^i_{sa}\}_{i=1}^N;\boldsymbol{b}_{sa},\gamma'''\right)\right)$$

$$\leq \sum_{a\in\mathcal{A}}-\alpha^\star_{sa,\gamma^\ell}\gamma^\ell+\frac{1}{N}\sum_{i=1}^{N}\mathfrak{D}_q(\hat{\boldsymbol{p}}^i_{sa},\boldsymbol{b}_{sa},\alpha^\star_{sa,\gamma^\ell})-\left(-\alpha^\star_{sa,\gamma^\ell}\gamma'''+\frac{1}{N}\sum_{i=1}^{N}\mathfrak{D}_q(\hat{\boldsymbol{p}}^i_{sa},\boldsymbol{b}_{sa},\alpha^\star_{sa,\gamma^\ell})\right)$$

$$= \sum_{a\in\mathcal{A}}\alpha^\star_{sa,\gamma^\ell}(\gamma'''-\gamma^\ell)$$

$$\leq (\gamma'''-\gamma^\ell)\frac{\theta^q}{\left(\max_{a\in\mathcal{A}}\max\{\boldsymbol{b}_{sa}\}-\underline{\gamma}\right)},$$

where the first step is due to $\gamma^\ell < \gamma^\star$ and definition of $\gamma'''$, the second step is from the definition of function $f$, the third step is from the definition of $\mathfrak{P}(\{\hat{\boldsymbol{p}}^i_{sa}\}_{i=1}^N;\boldsymbol{b}_{sa},\gamma)$ and $\alpha^\star_{sa,\gamma}$, and the last step is due to (17) and $\gamma^\ell \in (\gamma''',\gamma^\star)$. Notice that above inequality is true for all $\gamma^\ell \in (\gamma''',\gamma^\star)$, we could let $\gamma^\ell \to \gamma^\star$, which leads to

$$\theta^q - f\left(\gamma^\star - \frac{2\delta\left(\max_{a\in\mathcal{A}}\max\{\boldsymbol{b}_{sa}\}-\underline{\gamma}\right)}{\theta^q}\right) \leq -2\delta < -\delta.$$

Then (16) is the direct consequence of above inequality. This finishes the proof of statement, hence finishes the proof of the theorem. $\square$

*Proof of Theorem 4.5*. The Algorithm 1 is the direct consequence of the procedures in the content, except computing the slope, which has been explained at the end of the proof of Proposition 4.3.
For the time complexity, we can see that the bisection method on $\gamma$ and $\alpha$ uses complexity $\mathcal{O}(\log\epsilon_1^{-1}\log\epsilon_2^{-1})$. For the subproblem (9), which costs time complexity $h_q(S)$, we need to solve

**Algorithm 2:** Fast algorithm to solve (9) with $q = 1$

---

**Input:** Sorted $\boldsymbol{b}_{sa}$ with $b_{san_1} \geq b_{san_2} \geq \cdots \geq b_{san_S}$.
**Initialization:** $r \leftarrow \boldsymbol{b}_{sa}^\top \hat{\boldsymbol{p}}_{sa}^i$ and $\boldsymbol{p}_{sa}^i \leftarrow \hat{\boldsymbol{p}}_{sa}^i$.
**for** $k = 1, \ldots, S - 1$ **do**

    **if** $2\hat{p}_{san_k}^i + \alpha(b_{san_S} - b_{san_k})\hat{p}_{san_k}^i \geq 0$ **then** break
    **else**
        $r\mathrel{+}= 2\hat{p}_{san_k}^i + \alpha(b_{san_S} - b_{san_k})\hat{p}_{sas'}^i.$
        $p_{san_S}^i \mathrel{+}= p_{san_k}^i$ and $p_{san_k}^i = 0.$

**end**
**Result:** Optimal objective value $r$ and optimal solution $\boldsymbol{p}_{sa}^i$ of (9) with $q = 1$.

---

it $NA$ times. Besides, computing the upper bound claimed in Proposition 4.2 requires finding $\min\{\boldsymbol{b}_{sa}\}$ for each $a \in \mathcal{A}$, which is in time complexity $\mathcal{O}(AS)$. So we get the time complexity of Algorithm 1 is $\mathcal{O}\left(h_q(S)NA\log \epsilon_1^{-1} \log \epsilon_2^{-1} + AS\right)$.

$\square$

*Proof of Proposition 4.6.* Plug $\mathbb{B}_s^\infty$ in (3) into (4), we get the Bellman update is given by

$$\frac{1}{N}\max_{\boldsymbol{\pi}_s \in \Delta_A}\left\{\min \sum_{a \in \mathcal{A}}\sum_{i=1}^N \pi_{sa}\boldsymbol{p}_{sa}^i{}^\top(\boldsymbol{r}_{sa} + \lambda\boldsymbol{v}) \; : \; \left\|\boldsymbol{p}_{sa}^i - \hat{\boldsymbol{p}}_{sa}^i\right\|_\infty \leq \theta, \; \boldsymbol{p}_{sa}^i \in \Delta_S, \; \forall i \in [N], \; \forall a \in \mathcal{A}\right\}$$

$$= \frac{1}{N}\max_{\boldsymbol{\pi}_s \in \Delta_A}\sum_{a \in \mathcal{A}} \pi_{sa}\sum_{i=1}^N \min\left\{\boldsymbol{p}_{sa}^i{}^\top(\boldsymbol{r}_{sa} + \lambda\boldsymbol{v}) \; : \; \left\|\boldsymbol{p}_{sa}^i - \hat{\boldsymbol{p}}_{sa}^i\right\|_\infty \leq \theta, \; \boldsymbol{p}_{sa}^i \in \Delta_S\right\}$$

$$= \frac{1}{N}\max_{a \in \mathcal{A}}\sum_{i=1}^N \min_{\boldsymbol{p}_{sa}^i \in \Delta_S}\left\{\boldsymbol{b}_{sa}^\top\boldsymbol{p}_{sa}^i \; : \; \left\|\boldsymbol{p}_{sa}^i - \hat{\boldsymbol{p}}_{sa}^i\right\|_\infty \leq \theta\right\}.$$

The first equality is due to that the decision variables $\boldsymbol{p}_{sa}^i, \; \forall i \in [N]$ and $\forall a \in \mathcal{A}$, are independent from each other. Then we can divide the orignal problem into $NA$ subproblems. The second equality is from the fact that the objective function is affine w.r.t. $\pi_{sa}$, and the maximum is simply the greatest coefficient of $\pi_{sa}$. $\square$

## A.2 Proofs of results in Section 5

### A.2.1 Proof of results in Section 5.1

*Proof of Proposition 5.1.* We introduce the variable $v \geq \|\boldsymbol{p}_{sa}^i - \hat{\boldsymbol{p}}_{sa}^i\|_1$, so

$$\begin{aligned}
\mathfrak{D}_1(\hat{\boldsymbol{p}}_{sa}^i, \boldsymbol{b}_{sa}, \alpha) &= \min_{\boldsymbol{p}_{sa}^i \in \Delta_S}\|\boldsymbol{p}_{sa}^i - \hat{\boldsymbol{p}}_{sa}^i\|_1 + \alpha \cdot \boldsymbol{b}_{sa}^\top\boldsymbol{p}_{sa}^i \\
&= \min_{v \geq 0}\min_{\boldsymbol{p}_{sa}^i \in \Delta_S}\left\{v + \alpha\boldsymbol{b}_{sa}^\top\boldsymbol{p}_{sa}^i \; : \; \|\boldsymbol{p}_{sa}^i - \hat{\boldsymbol{p}}_{sa}^i\|_1 \leq v\right\} \\
&= \min_{v \geq 0}\left\{v + \alpha\min_{\boldsymbol{p}_{sa}^i \in \Delta_S}\boldsymbol{b}_{sa}^\top\boldsymbol{p}_{sa}^i \; : \; \|\boldsymbol{p}_{sa}^i - \hat{\boldsymbol{p}}_{sa}^i\|_1 \leq v\right\}.
\end{aligned} \tag{18}$$

$\square$

*Proof of Theorem 5.2.* We denote the objective function in (11) as

$$F(v) = v + \alpha\left\{\min_{\boldsymbol{p}_{sa}^i \in \Delta_S}\boldsymbol{b}_{sa}^\top\boldsymbol{p}_{sa}^i \; : \; \|\boldsymbol{p}_{sa}^i - \hat{\boldsymbol{p}}_{sa}^i\|_1 \leq v\right\}.$$

W.L.O.G., we assume that $b_{sa1} > b_{sa2} > \cdots > b_{saS}$, and $\hat{p}_{sa}^i > 0$. We claim that

$$
F(v) = \begin{cases}
v + \alpha \left( -\dfrac{v\,(b_{sa1} - b_{saS})}{2} + \boldsymbol{b}_{sa}^\top \hat{\boldsymbol{p}}_{sa}^i \right) & \text{if } v \in \left[0, 2\hat{p}_{sa1}^i\right), \\[2ex]
v + \alpha \displaystyle\sum_{k=K+1}^{S} r_k b_{sak} & \text{if } v \in \left[2\displaystyle\sum_{k=1}^{K} \hat{p}_{sak}^i, 2\displaystyle\sum_{k=1}^{K+1} \hat{p}_{sak}^i\right), \text{ for some } K \in [S-2], \\[3ex]
v + \alpha b_{saS} & \text{if } v \in \left[2\displaystyle\sum_{k=1}^{S-1} \hat{p}_{sak}^i, +\infty\right),
\end{cases}
$$

where

$$
r_k = \begin{cases}
\hat{p}_{sa(K+1)}^i - \left(v - 2\displaystyle\sum_{k=1}^{K} \hat{p}_{sak}^i\right)\Big/ 2 & \text{if } k = K+1, \\[2ex]
\hat{p}_{sak}^i & \text{if } K+2 \le k \le S-1, \\[1ex]
\hat{p}_{saS}^i + \dfrac{v}{2} & \text{if } k = S.
\end{cases}
$$

To prove the first case of the claim, it suffices to show that the optimal solution for the minimization problem in $F(v)$ is given by $\boldsymbol{p}^\star$, whose components are $p_1^\star = \hat{p}_{sa1}^i - \frac{v}{2}, p_k^\star = \hat{p}_{sak}^i, \forall 2 \le k \le S-1$ and $p_S^\star = \hat{p}_{saS}^i + \frac{v}{2}$. It can be easily verified that $\boldsymbol{p}^\star$ defined in this way satisfies the constraints of the minimization problem in $F(v)$.

To see the optimality, we consider any optimal solution $\bar{\boldsymbol{p}}$. We first notice that $\bar{p}_1 > 0$ and $\bar{p}_S < 1$. Actually, Let

$$
\begin{aligned}
\mathcal{N} &= \{k \in [S] \;:\; \bar{p}_k < \hat{p}_{sak}^i\}, \\
\mathcal{P} &= \{k \in [S] \;:\; \bar{p}_k > \hat{p}_{sak}^i\}, \\
\mathcal{E} &= \{k \in [S] \;:\; \bar{p}_k = \hat{p}_{sak}^i\}.
\end{aligned}
$$

Then by $\boldsymbol{e}^\top \bar{\boldsymbol{p}} = \boldsymbol{e}^\top \hat{\boldsymbol{p}}_{sa}^i = 1$ and $\|\bar{\boldsymbol{p}} - \hat{\boldsymbol{p}}_{sa}^i\|_1 \le v$, we get

$$
\sum_{k\in\mathcal{N}} \hat{p}_{sak}^i - \bar{p}_k = \sum_{k\in\mathcal{P}} \bar{p}_k - \hat{p}_{sak}^i
$$

$$
\sum_{k\in\mathcal{N}} (\hat{p}_{sak}^i - \bar{p}_k) + \sum_{k\in\mathcal{P}} (\bar{p}_k - \hat{p}_{sak}^i) \le v
$$

Hence $\sum_{k\in\mathcal{N}} \hat{p}_{sak}^i - \bar{p}_k = \sum_{k\in\mathcal{P}} \bar{p}_k - \hat{p}_{sak}^i \le \frac{v}{2} < \hat{p}_{sa1}^i$, so we get $\bar{p}_1 > 0$ and $\bar{p}_S < 1$.

Next we show that $\bar{p}_k = \hat{p}_{sak}^i, \forall 2 \le k \le S-1$. Otherwise we have some $2 \le \hat{k} \le S-1$ such that $|\bar{p}_{\hat{k}} - \hat{p}_{sa\hat{k}}^i| > 0$. If $\bar{p}_{\hat{k}} > \hat{p}_{sa\hat{k}}^i$, we define $\tilde{\boldsymbol{p}}$ with that $\tilde{p}_{\hat{k}} = \bar{p}_{\hat{k}} - \varepsilon$ and $\tilde{p}_S = \bar{p}_S + \varepsilon$, here $0 < \varepsilon < \min\{\frac{|\bar{p}_{\hat{k}} - \hat{p}_{sa\hat{k}}^i|}{2}, \frac{1-\bar{p}_S}{2}\}$, while keeping the other components of $\tilde{\boldsymbol{p}}$ same as $\bar{\boldsymbol{p}}$. We can see that $\tilde{\boldsymbol{p}}$ achieves smaller objective value than $\bar{\boldsymbol{p}}$ does due to $b_{sa\hat{k}} > b_{saS}$, which contradicts the optimality of $\bar{\boldsymbol{p}}$. If $\bar{p}_{\hat{k}} < \hat{p}_{sa\hat{k}}^i$, then we define $\tilde{\boldsymbol{p}}$ with that $\tilde{p}_{\hat{k}} = \bar{p}_{\hat{k}} + \varepsilon$ and $\tilde{p}_1 = \bar{p}_1 - \varepsilon$, here $0 < \varepsilon < \min\{\frac{|\bar{p}_{\hat{k}} - \hat{p}_{sa\hat{k}}^i|}{2}, \frac{\bar{p}_1}{2}\}$, while keeping the other components of $\tilde{\boldsymbol{p}}$ same as $\bar{\boldsymbol{p}}$. Similarly, $\tilde{\boldsymbol{p}}$ achieves smaller objective value and this implies the contradiction.

Finally, we verify the rest two components. We introduce the variables $d_1 = \bar{p}_1 - \hat{p}_{sa1}^i$ and $d_S = \bar{p}_S - \hat{p}_{saS}^i$, then we get the equivalent reformulation of the inner minimization in $F(v)$:

$$
\begin{aligned}
\min_{d_1, d_S} \quad & b_{sa1} d_1 + b_{saS} d_S \\
\text{s.t.} \quad & d_1 + d_S = 0, |d_1| + |d_S| \le v.
\end{aligned}
$$

The optimal $d_1$ and $d_S$ are given by $-v/2$ and $v/2$ respectively. Hence we proved $\boldsymbol{p}^\star$ is indeed an optimal solution.

To prove the second case of the claim, the optimal solution for the minimization problem in $F(v)$ is given by $\boldsymbol{p}^\star$, whose components are $p_k^\star = 0, \forall k \in [K]$, and $p_k^\star = r_k, \forall K+1 \le k \le S$. The decision variables $\boldsymbol{p}^\star$ defined in this way satisfies the constraints of the minimization problem in $F(v)$.

To see the optimality, we consider any optimal solution $\bar{\boldsymbol{p}}$. We first notice that $\bar{p}_{K+1} > 0$ and $\bar{p}_S < 1$.

To argue this by contradiction, we suppose the contrary $\bar{p}_{K+1} = 0$. Define the notations $\mathcal{N}, \mathcal{P}$ and $\mathcal{E}$ same as before. Then there exists $\hat{k} \leq K$ with $\bar{p}_{\hat{k}} > 0$, otherwise we assume that $\bar{p}_k = 0, \ \forall k \in [K]$, which implies the contradiction as follows.

$$\sum_{k=1}^{K+1} \hat{p}_{sak}^i > \frac{v}{2} \geq \sum_{k \in \mathcal{N}} \hat{p}_{sak}^i - \bar{p}_k \geq \sum_{k=1}^{K+1} \hat{p}_{sak}^i.$$

Here the first inequality is due to the selection of $v$, the second inequality has been deduced in the first case and the third inequality is from $\bar{p}_k = 0, \ \forall k \in [K+1]$. So we are able to find $\hat{k} \leq K$ with $\bar{p}_{\hat{k}} > 0$. By moving probability $\varepsilon = \min\{\frac{\bar{p}_{\hat{k}}}{2}, \frac{\hat{p}_{sa(K+1)}^i}{2}\}$ from $\bar{p}_{\hat{k}}$ to $\bar{p}_{K+1}$, we can achieve smaller objective value while keeping the feasibility, hence we get the contradiction, which implies that $\bar{p}_{K+1} > 0$ and $\bar{p}_S < 1$. By applying the similar procedures in the first case (moving some probability from $\bar{p}_{K+1}$ or to $\bar{p}_S$), we can show that $\bar{p}_k = \hat{p}_{sak}^i = r_k$ for $K+2 \leq k \leq S-1$.

Next we prove that $\bar{p}_k = r_k = 0$ for $k \in [K]$. Suppose the contrary is true; that is, $\bar{p}_{\hat{k}} > 0$ for some $\hat{k} \leq K$. Then we are able to apply the same procedures as before which illustrate that $\bar{p}_k = \hat{p}_{sak}^i, \ \forall \hat{k} < k < S$. Provided this, one can verified that the optimal strategy is putting all the rest probability $1 - \sum_{k=\hat{k}+1}^{S-1} \hat{p}_{sak}^i$ to $\bar{p}_S$ since $b_{sa1} > b_{sa2} > \cdots > b_{saS}$, and $v \geq 2\sum_{k=1}^{K} \hat{p}_{sak}^i \geq 2\sum_{k=1}^{\hat{k}} \hat{p}_{sak}^i$, which implies $\bar{p}_{sa\hat{k}} = 0$ and we get a contradiction. Hence $\bar{p}_k = r_k = 0, \ \forall k \in [K]$. Finally, we verify the rest two components. We introduce the variables $d_{K+1} = \bar{p}_{K+1} - \hat{p}_{sa(K+1)}^i$ and $d_S = \bar{p}_S - \hat{p}_{saS}^i$, then we get the equivalent reformulation of the inner minimization in $F(v)$:

$$\min_{d_{K+1}, d_S} \quad b_{sa(K+1)} d_{K+1} + b_{saS} d_S$$
$$\text{s.t.} \quad d_{K+1} + d_S = \sum_{k=1}^{K} \hat{p}_{sak}^i, \ |d_{K+1}| + |d_S| \leq v - \sum_{k=1}^{K} \hat{p}_{sak}^i.$$

The optimal $d_{K+1}$ and $d_S$ are given by $-\frac{v - 2\sum_{k=1}^{K} \hat{p}_{sak}^i}{2}$ and $\frac{v}{2}$ respectively. Hence we proved $\boldsymbol{p}^\star$ is an optimal solution.

To prove the third case of the claim, we notice that the optimal solution $\boldsymbol{e}_S$ to $\min_{\boldsymbol{p}_{sa}^i \in \Delta_S} \boldsymbol{b}_{sa}^\top \boldsymbol{p}_{sa}^i$ is also feasible for the inner minimization problem in $F(v)$, hence it becomes the optimal solution we desire, which implies $F(v) = v + \alpha b_{saS}$ at this time. This finishes the proof of our claim.

Our claim directly illustrates that $F(v)$ is a piecewise-linear function in $v$ with breakpoints $\{0\} \cup \{2\sum_{k=1}^{K} \hat{p}_{sak}^i : \forall K \in [S-1]\}$. Furthermore, based on the provided formulation of $F(v)$, we can compute the difference of value for $F(\cdot)$ between any two adjacent breakpoints, given by

$$F(v_K) - F(v_{K-1}) = 2\hat{p}_{saK}^i + \alpha(b_{saS} - b_{saK})\hat{p}_{saK}^i \quad \forall K \in [S-1],$$

where $v_K = 2\sum_{k=1}^{K} \hat{p}_{sak}^i, \ \forall K \in [S-1]$ and $v_0 = 0$.

Hence we provide Algorithm 2 to compute (11), whose time complexity is $\mathcal{O}(S \log S)$ generally and can be reduced to $\mathcal{O}(S)$ if the sorted $\boldsymbol{b}_{sa}$ is provided.

$\square$

*Proof of Corollary 5.2.1.* We can see that Algorithm 2 is in time complexity $\mathcal{O}(S)$ if $\cup_{a \in \mathcal{A}} \boldsymbol{b}_{sa}$ are sorted, which can be done at the Initialization step in Algorithm 1 with time complexity $\mathcal{O}(AS \log S)$. So by Theorem 4.5, we get the overall complexity is $\mathcal{O}\left(NAS \log \epsilon_1^{-1} \log \epsilon_2^{-1} + AS \log S\right)$.

$\square$

**Algorithm 3:** Fast algorithm to solve the inner minimization problem in (10)

---

**Input:** $v$, $r_{sa}$, and $\hat{p}_{sa}^i \in \Delta_S$.
**Initialization:** $p_{sa}^{i,\star} = \mathbf{0}$.
Sort $b_{sa}$ as $b_{san_1} \le \cdots \le b_{san_S}$.
Find the smallest $k$ such that $\sum_{j=1}^k \left( \hat{p}_{san_j}^i + \theta \right) \ge 1$.
Set $p_{san_j}^{i,\star} = \hat{p}_{san_j}^i + \theta$ for $j \le k-1$ and $p_{san_k}^{i,\star} = 1 - \sum_{j=1}^{k-1} \left( \hat{p}_{san_j}^i + \theta \right)$.
$r = b_{sa}^\top p_{sa}^{i,\star}$.
**Result:** Optimal objective value $r$ and optimal solution $p_{sa}^\star$ of the inner minimization problem in (10).

---

### A.2.2 Proof of results in Section 5.2

*Proof of Theorem 5.3.* From problem (9), we can get the minimization problem for $q = 2$

$$
\begin{aligned}
& \min_{p_{sa}^i \in \Delta_S} \left\| p_{sa}^i - \hat{p}_{sa}^i \right\|_2^2 + \alpha b_{sa}^\top p_{sa}^i \\
=\ & \min_{p_{sa}^i \in \Delta_S} \left\| p_{sa}^i \right\|_2^2 - 2 p_{sa}^{i\top} \hat{p}_{sa}^i + \left\| \hat{p}_{sa}^i \right\|_2^2 + \alpha b_{sa}^\top p_{sa}^i \\
=\ & \min_{p_{sa}^i \in \Delta_S} \left\| p_{sa}^i \right\|_2^2 - \left( 2\hat{p}_{sa}^i - \alpha b_{sa} \right)^\top p_{sa}^i + \left\| \hat{p}_{sa}^i \right\|_2^2 \\
=\ & -\frac{\alpha^2 \left\| b_{sa} \right\|_2^2}{4} + \alpha b_{sa}^\top \hat{p}_{sa}^i + \min_{p_{sa}^i \in \Delta_S} \left\| p_{sa}^i - \frac{2\hat{p}_{sa}^i - \alpha b_{sa}}{2} \right\|_2^2 .
\end{aligned}
$$

So it suffices to solve $\min_{p_{sa}^i \in \Delta_S} \left\| p_{sa}^i - \frac{2\hat{p}_{sa}^i - \alpha b_{sa}}{2} \right\|_2^2$, which can be done by Euclidean projection algorithm (Wang and Carreira-Perpinán, 2013) with time complexity $\mathcal{O}(S \log S)$. $\qquad\square$

*Proof of Corollary 5.3.1.* The result is the direct consequence of Theorem 4.5 with $h_2(S)$ is $\mathcal{O}(S \log S)$, provided by Theorem 5.3.

$\qquad\square$

### A.2.3 Proof of results in Section 5.3

*Proof of Theorem 5.4.* We claim that the Algorithm 3 solves problem (13) with time complexity $\mathcal{O}(S \log S)$. By expanding the $\infty$-norm, we formulate (13) as the following box constraints problem.

$$
\begin{aligned}
\min\ & b_{sa}^\top p_{sa}^i \\
\text{s.t.}\ & \max \left\{ \mathbf{0}, \hat{p}_{sa}^i - \theta e \right\} \le p_{sa}^i \le \min \left\{ e, \hat{p}_{sa}^i + \theta e \right\}, \\
& e^\top p_{sa}^i = 1, \\
& p_{sa}^i \in \mathbb{R}^S .
\end{aligned}
$$

To get the optimal solution, we put the probability on the index where $b_{sa}$ is small as much as possible. Specifically, we assume $b_{sa1} < b_{sa2} < \cdots < b_{saS}$ w.l.o.g., and assume $k$ is the smallest index such that $\sum_{j=1}^k \left( \hat{p}_{saj}^i + \theta \right) \ge 1$. We claim that

$$
p_{saj}^{i,\star} = \begin{cases}
\hat{p}_{saj}^i + \theta & \text{if } 1 \le j \le k-1 \\
1 - \displaystyle\sum_{\ell=1}^{k-1} \left( \hat{p}_{sa\ell}^i + \theta \right) & \text{if } j = k \\
0 & \text{otherwise} .
\end{cases}
$$

is the optimal solution to the above formulation. To see this, suppose $\bar{p}$ is optimal and there is some $\hat{j} \in [k-1]$ with $\bar{p}_{\hat{j}} \ne \hat{p}_{saj}^i + \theta$. By above box constraint, we get $\bar{p}_{\hat{j}} < \hat{p}_{sa\hat{j}}^i + \theta$, so there exists $\bar{j} \ge k$ such that $\bar{p}_{\bar{j}} > p_{saj}^{i,\star}$. By moving the probability from $\bar{p}_{\bar{j}}$ to $\bar{p}_{\hat{j}}$, we can achieve strictly smaller objective value, which is a contradiction with that $\bar{p}$ is optimal. This gives us an optimal solution with the first $k-1$ components coincides $p_{sa}^{i,\star}$. Then we can get $p_{sa}^{i,\star}$ is indeed optimal by putting the

extra $1 - \sum_{\ell=1}^{k-1} \left( \hat{p}_{sa\ell}^i + \theta \right)$ probability on $p_{sak}^{i,\star}$, since $b_{sak} \leq \cdots \leq b_{saS}$.

The major time complexity of Algorithm 3 is sorting the vector $\boldsymbol{b}_{sa} \in \mathbb{R}^S$, which is $\mathcal{O}(S \log S)$.

$\qquad\square$

*Proof of Corollary 5.4.1.* As for each $a \in \mathcal{A}$, we need to sort $\boldsymbol{b}_{sa}$, which costs $\mathcal{O}(AS \log S)$, then we need to solve $NA$ subproblems, which costs $\mathcal{O}(NAS)$, so the whole problem is computed in time $\mathcal{O}(AS \log S + NAS)$.

$\qquad\square$

## B  Appendix: Computational Complexity for General Convex Optimzation

To compare our algorithm with general convex optimization algorithm, we use general convex optimization to compute the problem (5) and problem (8). The time complexities will be discussed in different situations:

- Suppose $q = 1$: For general convex optimization problem, problem (5) is equivalent with the following problem:

$$
[\mathfrak{T}(\boldsymbol{v})]_s = 
\left[
\begin{array}{ll}
\text{minimize} & \gamma \\
\text{subject to} & \dfrac{1}{N} \sum_{i=1}^{N} (\boldsymbol{r}_{sa} + \lambda \boldsymbol{v})^\top \boldsymbol{p}_{sa}^i \leq \gamma, \ \forall a \in \mathcal{A} \\
& \dfrac{1}{N} \sum_{i=1}^{N} \sum_{a \in \mathcal{A}} \sum_{s' \in \mathcal{S}} \left| p_{sas'}^i - \hat{p}_{sas'}^i \right| \leq \theta \\
& \gamma \in \mathbb{R}, \ \boldsymbol{p}_{sa}^i \in \Delta_S, \ \forall i \in [N], \ \forall a \in \mathcal{A}.
\end{array}
\right]
\quad \forall s \in \mathcal{S}.
$$

By introducing the variables $t_{sas'}^i = \left| p_{sas'}^i - \hat{p}_{sas'}^i \right|$, $\forall i \in [N], \ \forall a \in \mathcal{A}, \ \forall s' \in \mathcal{S}$, the above problem is equivalent with

$$
\begin{aligned}
\text{minimize} \quad & \gamma \\
\text{subject to} \quad & \frac{1}{N} \sum_{i=1}^{N} (\boldsymbol{r}_{sa} + \lambda \boldsymbol{v})^\top \boldsymbol{p}_{sa}^i \leq \gamma, \ \forall a \in \mathcal{A} \\
& \frac{1}{N} \sum_{i=1}^{N} \sum_{a \in \mathcal{A}} \sum_{s' \in \mathcal{S}} t_{sas'}^i \leq \theta \\
& t_{sas'}^i \geq p_{sas'}^i - \hat{p}_{sas'}^i, \ t_{sas'}^i \geq \hat{p}_{sas'}^i - p_{sas'}^i, \ \forall i \in [N], \ \forall a \in \mathcal{A}, \ \forall s' \in \mathcal{S} \\
& \gamma \in \mathbb{R}, \ \boldsymbol{p}_{sa}^i \in \Delta_S, \ t_{sas'}^i \in \mathbb{R}, \ \forall i \in [N], \ \forall a \in \mathcal{A}, \ \forall s' \in \mathcal{S}.
\end{aligned}
$$

There are $1 + NSA + NSA = \mathcal{O}(NSA)$ decision variables, and the number of bits in the input is $\mathcal{O}(1) + \mathcal{O}(NAS) + \mathcal{O}(NAS) + \mathcal{O}(NAS) + \mathcal{O}(NAS) + \mathcal{O}(NAS) = \mathcal{O}(NAS)$. So by (Karmarkar, 1984), the complexity of solving this LP is $\mathcal{O}(N^{4.5} S^{4.5} A^{4.5})$.

We utilize our outer bisection, solving (6) directly using convex optimization. Typically, for each fixed $a \in \mathcal{A}$ and $\gamma$, (6) is equivalent with the LP that

$$
\begin{aligned}
\text{minimize} \quad & \frac{1}{N} \sum_{i=1}^{N} \sum_{s' \in \mathcal{S}} t_{sas'}^i \\
\text{subject to} \quad & \frac{1}{N} \sum_{i=1}^{N} \boldsymbol{b}_{sa}^\top \boldsymbol{p}_{sa}^i \leq \gamma \\
& t_{sas'}^i \geq p_{sas'}^i - \hat{p}_{sas'}^i, \ t_{sas'}^i \geq \hat{p}_{sas'}^i - p_{sas'}^i, \ \forall i \in [N], \ \forall s' \in \mathcal{S} \\
& \boldsymbol{p}_{sa}^i \in \Delta_S, \ t_{sas'}^i \in \mathbb{R}, \ \forall i \in [N], \ \forall s' \in \mathcal{S}.
\end{aligned}
$$

There are $2NS$ decision variables, and the number of bits in the input is $\mathcal{O}(NS + NS + NS + NS) = \mathcal{O}(NS)$. So the complexity of solving this LP is $\mathcal{O}(N^{4.5} S^{4.5})$. Together with the outer bisection, the total complexity for each Bellman update with $\epsilon_1$ tolerance is $\mathcal{O}(N^{4.5} S^{4.5} A \log \epsilon_1^{-1})$.

We utilize our nested bisection scheme, solving (9) using the general convex optimization algorithm. Typically, for each fixed $a \in \mathcal{A}, i \in [N]$ and $\alpha$, (9) is equivalent with

$$
\begin{aligned}
\text{minimize} \quad & \sum_{s' \in \mathcal{S}} t^i_{sas'} + \alpha \cdot \boldsymbol{b}^\top_{sa} \boldsymbol{p}^i_{sa} \\
\text{subject to} \quad & t^i_{sas'} \geq p^i_{sas'} - \hat{p}^i_{sas'}, \; t^i_{sas'} \geq \hat{p}^i_{sas'} - p^i_{sas'}, \; \forall s' \in \mathcal{S} \\
& \boldsymbol{p}^i_{sa} \in \Delta_S, \; t^i_{sas'} \in \mathbb{R}, \; \forall s' \in \mathcal{S}.
\end{aligned}
$$

There are $2S$ decision variables, and the number of bits in the input is $\mathcal{O}(S+S+S) = \mathcal{O}(S)$. So the complexity of solving this LP is $\mathcal{O}(S^{4.5})$. Together with the nested bisection, the total complexity for each Bellman update where the tolerances of bisections are $\epsilon_1$ and $\epsilon_2$, is $\mathcal{O}(NS^{4.5}A \log \epsilon_1^{-1} \log \epsilon_2^{-1})$.

- Suppose $q = 2$: For general convex optimization problem, problem (5) is equivalent with the following SOCP:

$$
[\mathfrak{T}(\boldsymbol{v})]_s = \left[ \begin{array}{ll}
\text{minimize} & \gamma \\
\text{subject to} & \dfrac{1}{N} \sum_{i=1}^{N} (\boldsymbol{r}_{sa} + \lambda \boldsymbol{v})^\top \boldsymbol{p}^i_{sa} \leq \gamma, \; \forall a \in \mathcal{A} \\
& \dfrac{1}{N} \sum_{i=1}^{N} \sum_{a \in \mathcal{A}} \left\| \boldsymbol{p}^i_{sa} - \hat{\boldsymbol{p}}^i_{sa} \right\|^2_2 \leq \theta^2 \\
& \gamma \in \mathbb{R}, \; \boldsymbol{p}^i_{sa} \in \Delta_S, \; \forall i \in [N], \; \forall a \in \mathcal{A}.
\end{array} \right] \quad \forall s \in \mathcal{S}.
$$

There are $1 + NSA = \mathcal{O}(NSA)$ decision variables, and the number of constraints are $A + 1 + NA(S+2) = \mathcal{O}(NAS)$. So the complexity of solving the SOCP with $\epsilon$-accuracy is $\mathcal{O}(\sqrt{NAS} \log \epsilon^{-1} \cdot (NSA)^2 (NAS + 1 + 2(A + NA(S+2)))) = \mathcal{O}(N^{3.5} S^{3.5} A^{3.5} \log \epsilon^{-1})$.

We utilize our outer bisection, solving (6) directly using convex optimization. Typically, for each fixed $a \in \mathcal{A}$ and $\gamma$, (6) is equivalent with the SOCP that

$$
\begin{aligned}
\text{minimize} \quad & \delta \\
\text{subject to} \quad & \frac{1}{N} \sum_{i=1}^{N} \boldsymbol{b}^\top_{sa} \boldsymbol{p}^i_{sa} \leq \gamma \\
& \frac{1}{N} \sum_{i=1}^{N} \left\| \boldsymbol{p}^i_{sa} - \hat{\boldsymbol{p}}^i_{sa} \right\|^2_2 \leq \delta \\
& \delta \in \mathbb{R}, \; \boldsymbol{p}^i_{sa} \in \Delta_S, \; \forall i \in [N].
\end{aligned}
$$

There are $1 + NS = \mathcal{O}(NS)$ decision variables, and the number of constraints are $\mathcal{O}(NS)$. So the complexity of solving the above SOCP with $\epsilon$-accuracy is $\mathcal{O}(\sqrt{NS} \log \epsilon^{-1} \cdot N^2 S^2 (NS + 1 + 2(1 + N(S+2)))) = \mathcal{O}(N^{3.5} S^{3.5} \log \epsilon^{-1})$, hence the total complexity of the Bellman update is $\mathcal{O}(N^{3.5} S^{3.5} A \log \epsilon^{-1} \log \epsilon_1^{-1})$.

We utilize our nested bisection scheme, solving problem (9) using the general convex optimization algorithm. For each fixed $a \in \mathcal{A}, i \in [N]$ and $\alpha$, problem (9) is equivalent with

$$
\begin{aligned}
\text{minimize} \quad & \delta + \alpha \cdot \boldsymbol{b}^\top_{sa} \boldsymbol{p}^i_{sa} \\
\text{subject to} \quad & \left\| \boldsymbol{p}^i_{sa} - \hat{\boldsymbol{p}}^i_{sa} \right\|^2_2 \leq \delta \\
& \boldsymbol{p}^i_{sa} \in \Delta_S, \; \delta \in \mathbb{R}.
\end{aligned}
$$

There are $1 + S = \mathcal{O}(S)$ decision variables, and the number of constraints are $\mathcal{O}(S)$. So the complexity of solving the above SOCP with $\epsilon$-accuracy is $\mathcal{O}(\sqrt{S} \log \epsilon^{-1} \cdot S^2 (S + 1 + 2(1 + S))) = \mathcal{O}(S^{3.5} \log \epsilon^{-1})$, hence the total complexity of the Bellman update is $\mathcal{O}(NS^{3.5} A \log \epsilon^{-1} \log \epsilon_1^{-1} \log \epsilon_2^{-1})$.

- Suppose $q = \infty$: We can also consider solving the inner problem in (10) using general convex optimization, which is equivalent with

$$
\begin{aligned}
\text{minimize} \quad & \boldsymbol{b}^\top_{sa} \boldsymbol{p}^i_{sa} \\
\text{subject to} \quad & p^i_{sas'} - \hat{p}^i_{sas'} \leq \theta, \; \hat{p}^i_{sas'} - p^i_{sas'} \leq \theta, \; \forall s' \in \mathcal{S} \\
& \boldsymbol{p}^i_{sa} \in \Delta_S,
\end{aligned}
$$

Table 1: Comparisons of runtime (average and standard deviation)(second) of Bellman updates for all algorithms in $L_1$ norm.

| Algorithm $\diagdown$ $N=S=A$ | 10 | 20 | 30 | 40 |
|---|---|---|---|---|
| Fast | 0.0253 (0.02) | 0.1243 (0.02) | 0.2794 (0.02) | 0.5112 (0.02) |
| Gurobi | 0.039 (0.02) | 0.7084 (0.03) | 4.3617 (0.05) | 22.6013 (0.92) |
| FOM(3 its) | 0.644 (1.0193) | 4.0821 (1.05) | 12.6608 (1.00) | 29.22 (1.18) |

Table 2: Comparisons of runtime (average and standard deviation)(second) of Bellman updates for all algorithms in $L_2$ norm.

| Algorithm $\diagdown$ $N=S=A$ | 50 | 60 | 70 | 80 |
|---|---|---|---|---|
| Fast | 7.3717 (0.11) | 10.8293 (0.12) | 16.8888 (0.12) | 24.6080 (0.07) |
| Gurobi | 5.1669 (0.09) | 11.6760 (0.14) | 23.6852 (0.34) | 44.3305 (0.18) |
| FOM(3 its) | 9.1936 (1.83) | 14.5653 (2.01) | 21.3651 (1.97) | 31.1326 (1.79) |

where $i \in [N]$ and $a \in \mathcal{A}$ are fixed.

There are $S$ decision variables, and the number of bits in the input is $\mathcal{O}(S + S + S) = \mathcal{O}(S)$. So the complexity of solving this LP is $\mathcal{O}(S^{4.5})$. Then the total complexity for each Bellman update is $\mathcal{O}(NS^{4.5}A)$.

## C Appendix: Details for Numerical Experiments

We compare our fast algorithm with the state-of-the-art solver Gurobi with version v10.0.1rc0 (Gurobi Optimization, LLC, 2023) and the first-order method of (Grand-Clément and Kroer, 2021a). All experiments are implemented in Python 3.8, and they are run on a 2.3 GHz 4-Core Intel Core i7 CPU with 32 GB 3733 MHz DDR4 main memory. We will release our code to ensure reproducibility. https://github.com/Chill-zd/Fast-Bellman-Updates-DRMDP

Our algorithms are tested on some random instances generated by the Generalized Average Reward Non-stationary Environment Test-bench (Garnet MDPs) (Archibald et al., 1995; Bhatnagar et al., 2009). The Garnet MDPs are a collection of assessment problems designed to assess the performance of reinforcement learning algorithms in non-stationary environments. It is convenient to construct and implement these problems. We utilize the parameter $n_b$ to regulate the proportion of the next states accessible for each state-action pair $(s, a)$. Following the same setting as (Grand-Clément and Kroer, 2021a), we set $n_b$ to be 0.2 and random uniform rewards to be in $[0, 10]$. The discount factor $\lambda$ is fixed at 0.8, and parameter $\epsilon = 0.1$. For each MDP instance, we generate the sampled kernels $\boldsymbol{p}^1, \ldots, \boldsymbol{p}^N$, considering $N$ small random (Garnet) perturbations around the nominal kernel $\boldsymbol{p}^0$. We set parameter $\theta$ in Proposition 4.1 to be $\sqrt{n_b A}$.

To test the speed of Bellman update, we run the random instances 50 times for all the algorithms, and show the average time of them in tables 1, 2 and 3. We can see that our algorithm performs better than Gurobi and the first-order method. When the states number increases, the running time of our algorithm keeps a small standard deviation.

We also compare the speed of value iteration for all algorithms using the same convergence criteria: $\|\boldsymbol{v} - \boldsymbol{v}^\star\|_\infty \leq 2\lambda\epsilon(1 - \lambda)^{-1}$, which follows (Grand-Clément and Kroer, 2021a). The results are shown in tables 4, 5 and 6. The runtimes that exceed 4000s for $L_1$ and $L_\infty$ case or exceed 10000s for

Table 3: Comparisons of runtime (average and standard deviation) (millisecond) of Bellman updates for all algorithms in $L_\infty$ norm.

| Algorithm $\diagdown$ $N=S=A$ | 10 | 20 | 30 | 40 |
|---|---|---|---|---|
| Fast | 0.06 (0.1) | 0.21 (0.1) | 0.47 (0.2) | 0.80 (0.4) |
| Gurobi | 0.92 (0.1) | 7.33 (0.8) | 26.50 (3.8) | 65.22 (7.9) |
| FOM | 144.92 (981.1) | 166.19 (966.5) | 237.24 (978.2) | 366.74 (976.6) |

$L_2$ case will be shown as "−". We can see from tables 4, 5 and 6 that our algorithm always performs better than Gurobi and the first-order method. We point out that our algorithm generally becomes better as the state number increases.

Table 4: Runtime (second) of value iteration for all algorithms in $L_1$ norm.

| $N = S = A$ / Algorithm | 10 | 20 | 30 | 40 |
|---|---|---|---|---|
| Fast | 3.6164 | 34.0930 | 116.4698 | 281.9007 |
| Gurobi | 5.7131 | 202.0723 | 1967.9552 | − |
| FOM | 202.4039 | 3578.1377 | − | − |

Table 5: Runtime (second) of value iteration for all algorithms in $L_2$ norm.

| $N = S = A$ / Algorithm | 40 | 50 | 60 | 70 |
|---|---|---|---|---|
| Fast | 2198.5413 | 4847.4994 | 7223.7663 | 12387.5601 |
| Gurobi | 2543.9588 | 4861.4114 | 7949.4562 | 22096.7578 |
| FOM | − | − | − | − |

Table 6: Runtime (second) of value iteration for all algorithms in $L_\infty$ norm.

| $N = S = A$ / Algorithm | 10 | 20 | 30 | 40 |
|---|---|---|---|---|
| Fast | 0.0045 | 0.0348 | 0.1371 | 0.3181 |
| Gurobi | 0.0907 | 0.8746 | 3.9243 | 11.5589 |
| FOM | 16.4344 | 97.44293 | 1131.5260 | 3933.8585 |

