# OpenReview forum: "Fast Bellman Updates for Wasserstein Distributionally Robust MDPs"
_NeurIPS.cc/2023/Conference — NeurIPS 2023 poster_

### Official Review · Reviewer_zZNP · 2023-06-08

**Soundness:** 3 good
**Presentation:** 3 good
**Contribution:** 3 good
**Rating:** 7
**Confidence:** 4

**Summary:**

* tailored algorithms for solving Wasserstein distributionally robust MDPs and fast implementations with $L_1$, $L_2$, $L_\infty$-based Wasserstein distance.

**Strengths:**

* Fastest algorithms (in terms of dependency on N, S, and A - the number of kernels/states/actions) for solving Wasserstein distributionally robust MDPs.
* Paper is easy to follow, short and straight-to-the-point.

**Weaknesses:**

* This is a rather niche application of principles that have been developed in several other papers for robust MDPs [1,2,3].

[1] Chin Pang Ho, Marek Petrik, and Wolfram Wiesemann. Fast bellman updates for robust mdps. In  International Conference on Machine Learning, pages 1979–1988. PMLR, 2018.

[2] Chin Pang Ho, Marek Petrik, and Wolfram Wiesemann. Partial policy iteration for l1-robust markov
 decision processes. The Journal of Machine Learning Research, 22(1):12612–12657, 2021.

[3]  Chin Pang Ho, Marek Petrik, and Wolfram Wiesemann. Robust phi-divergence mdps. arXiv preprint
 arXiv:2205.14202, 2022.

**Questions:**

Model : can your algorithms be extended to handle the case of both transitions and reward uncertainty?

Theorem 4.4: how do the $\epsilon_1$ and $\epsilon_2$ error propagate into the overall error of $v_t$, the t-th iterate of value iteration?

Wasserstein balls: how is the radius $\theta$ chosen in practice, given some dataset?

**Limitations:**

I do not see any serious limitations in this work. The only downside is the somewhat narrow scope in terms of outreach.

---

> ### Author Rebuttal · Authors · 2023-08-07
>
> Thank you very much for your positive comments and for taking the time to read our manuscript!
>
> **Weaknesses**
>
> 1. This is a rather niche application of principles that have been developed in several other papers for robust MDPs.
>
> In the same spirit of using first order methods for solving both the robust MDPs (RMDPs) and the distributionally robust MDPs (DRMDPs) [iv-v], while we agree that the proposed algorithm belongs to the class of bisection methods that solve RMDPs [i-iii], we would like to highlight the differences between our work and [i-iii]. (a) Firstly, we consider DRMDPs, which is a generalization of RMDPs, and so we are different from [i-iii] in terms of problem settings. (b) Secondly, the resemblance of our bisection step and [i-iii] only occurs in lines 200 - 213, which is less than 1/2 page in this paper. In particular, the second bisection step in the proposed nested bisection method and Proposition 4.3 are novel in this paper; without them, it is not easy to design algorithm(s) that have time complexity that is almost linear in the number of expected kernels, which do not exist in RMDPs [i-iii]. In our opinion, this is one of the key factors that our proposed algorithm outperforms state-of-the-art algorithms, such as [v]. (c) Finally, the subproblems, such as (11), are different from the case of RMDPs, and so it requires new theoretical results and algorithms for fast computations. Thank you for your question and we will clarify this point in our next version of manuscript.
>
> **Questions**
>
> 1. Model : can your algorithms be extended to handle the case of both transitions and reward uncertainty?
>
> Thank you for your great question. Yes, our algorithm can be extended to reward uncertainty, because the decomposition technique can be seamlessly applied to this extension. We will clarify this point in our next version of the manuscript.
>
> 2. Theorem 4.4: how do the $\epsilon_1$ and $\epsilon_2$ error propagate into the overall error of $v_t$, the t-th iterate of value iteration?
>
> Thanks again for your insightful question. We have proved a new result on the error bound for the Bellman update, which is $\mathcal{O}(\epsilon_1+\epsilon_2)$. Since the complexities of the proposed algorithms are in $\mathcal{O}(\log\epsilon_1^{-1}\log\epsilon_2^{-1})$, one can compute a highly accurate solution with very small $\epsilon_1$ and $\epsilon_2$. We will provide this result in our next version of the manuscript.
>
> 3. Wasserstein balls: how is the radius $\theta$ chosen in practice, given some dataset?
>
> In practice, $\theta$ is often chosen via cross-validation, e.g. [vi]. However, one could also derive theoretical bounds such that the ambiguity set contains the unknown true distribution with high confidence [vii].
>
> [i] Fast Bellman updates for robust MDPs. 2018.
>
> [ii] Partial policy iteration for l1-robust Markov decision processes. 2021.
>
> [iii] Robust phi-divergence MDPs. 2022.
>
> [iv] Scalable first-order methods for robust MDPs. 2021.
>
> [v] First-order methods for Wasserstein distributionally robust MDP. 2021.
>
> [vi] Distributionally robust inverse covariance estimation: the Wasserstein shrinkage estimator, 2020.
>
> [vii] Data-driven distributionally robust optimization using the Wasserstein metric: Performance guarantees and tractable reformulations. 2018.

---

> > ### Comment · Reviewer_zZNP · 2023-08-11
> >
> > Thank you for your reply. In light of the responses of the authors I am more convinced of the importance of the results in the paper and I have increased my score.

---

> > > ### Author Response · Authors · 2023-08-14
> > > **Thank you!!**
> > >
> > > Thank you very much for your time reading our rebuttal! Thanks a lot for being very supportive and recognizing our work!

---

### Official Review · Reviewer_JxmA · 2023-07-04

**Soundness:** 3 good
**Presentation:** 3 good
**Contribution:** 3 good
**Rating:** 6
**Confidence:** 2

**Summary:**

 - The paper proposes a computationally efficient solution framework to solve the distributionally robust Bellman operator induced by Wasserstein ambiguity sets, which is critical in performing distributionally robust value iteration algorithms.
- The proposed framework features a novel decomposition of the optimization problem involved in DR Bellman updates, based on which the updates are reduced to solving small subproblems.
- The paper shows the overall complexity of the proposed framework is quasi-linear in $S$ and $A$ when considering Wasserstein distance with $L_1$, $L_2$, or $L_\infty$ norm, proving the advantage of the proposed framework. The theory is further supported by numerical experiments.


**Strengths:**

- The problem of solving distributionally robust Bellman operator in a computationally efficient manner is a critical problem in DRMDP literatures. The presentation of the paper is also clear.

**Weaknesses:**

-  The framework seems to rely on the specific structure of the Wasserstein ambiguity set with specific reference distribution of the Wasserstein ball, which makes the application of the framework limited.

**Questions:**

- Regarding the formulation of distributionally robust MDPs, the reference distribution of the ambiguity set $\nu_s$ is defined as $\mu_s(\cdot) = \frac{1}{N}\sum_{i=1}^N\delta_{\hat{\boldsymbol{p}}_s^i}(\cdot)$ with $\hat{\boldsymbol{p}}^i$ being $N$ empirical transition kernels. What do these empirical distributions corresponds to in the real-world application of reinforcement learning algorithms? Typically, we have a single empirical transition kernel estimated from historical data. How can we interpret these $N$ empirical transition kernels?

**Limitations:**

The limitation is stated in previous parts.

---

> ### Author Rebuttal · Authors · 2023-08-07
>
> Thank you very much for your positive comments and your time to review our paper!
>
> **Weaknesses**
>
> 1. The framework seems to rely on the specific structure of the Wasserstein ambiguity set with the specific reference distribution of the Wasserstein ball, which makes the application of the framework limited.
>
> We would like to clarify that Wasserstein ambiguity sets have many nice properties such as consistency in optimality and finite-sample bounds [i], and thus provides a general and powerful metric for distributionally robust MDPs. In recent years, Wasserstein ambiguity sets are widely used in stochastic optimization [iv] and multistage stochastic optimization [v]. Compared to the $\phi$-divergence, the Wasserstein structure does not suffer from the absolutely continuous restriction and addresses how close two points in the support are to each other. More advantages of the Wasserstein structure are provided in [iii].
>
> **Questions**
>
> 1. Regarding the formulation of distributionally robust MDPs, the reference distribution of the ambiguity set is defined with $N$ empirical transition kernels. What do these empirical distributions correspond to in the real-world application of reinforcement learning algorithms? Typically, we have a single empirical transition kernel estimated from historical data. How can we interpret these $N$ empirical transition kernels?
>
> Thank you for your very insightful question! You are absolutely correct; the common practice is to estimate just one empirical transition kernel. However, in our context, one could/should consider the Bayesian point of view (instead of frequentist), see e.g. [vi]; That is, one could use historical data to estimate the posterior distribution of the transition kernel. Then the $N$ empirical transition kernels could be sampled from the posterior distribution.
>
>
> [i] PM Esfahani and D Kuhn. Data-driven distributionally robust optimization using the Wasserstein metric: Performance guarantees and tractable reformulations. 2018.
>
> [ii] I. Yang. A convex optimization approach to distributionally robust Markov decision processes with Wasserstein distance. 2017.
>
> [iii] R. Gao, A. Kleywegt. Distributionally robust stochastic optimization with Wasserstein distance. 2022.
>
> [iv] D. Wozabal. Robustifying convex risk measures for linear portfolios: A nonparametric approach. 2014.
>
> [v] G.C. Pflug and A. Pichler. Multistage stochastic optimization. 2014.
>
> [vi] R. H. Russel, M Petrik. Beyond confidence regions: tight Bayesian ambiguity sets for robust MDPs. 2019.

---

> > ### Comment · Reviewer_JxmA · 2023-08-16
> >
> > Thanks for answering my questions. I have read the rebuttals and the comments from other reviewers. I have no further concerns and I would raise my score to 6.

---

> > > ### Author Response · Authors · 2023-08-16
> > > **Thank you!!**
> > >
> > > Thank you very much for your time reading our rebuttal and supporting our work!!

---

### Official Review · Reviewer_SCaM · 2023-07-05

**Soundness:** 3 good
**Presentation:** 3 good
**Contribution:** 2 fair
**Rating:** 5
**Confidence:** 4

**Summary:**

The paper studies Wasserstein Distributionally robust MDPs (WDRMDP) problem when the ambiguity set is defined based on Wasserstein distance and rectangular. It is then well known that the optimal policy can be computed by solving Bellman equations, which have the form of distributionally robust linear programs. The authors then focus on solving the Bellman updates. The main idea is to transform the robust linear problem into parameterized problems so that the Bellman updates can be done via bisection. The paper then provides some new algorithms running in polynomial time complexity and claims that their complexities are smaller than those from prior works. Experiments show that their algorithm performs better than some baselines.

**Strengths:**

The problem under investigation holds significant importance. Over the past few decades, robust and distributionally robust Markov Decision Processes (MDPs) have garnered considerable attention. Resolving robust MDPs typically entails addressing computationally expensive minimax problems, necessitating the development of efficient algorithms. In this regard, the paper appears to succeed by presenting novel and efficient algorithms for solving the problem. The proposed algorithms exhibit sound and good time complexities. The experimental results seem convincing, effectively showcasing the efficiency of the new algorithm in comparison to prior algorithms.

**Weaknesses:**

The WDRMDP problem itself is not novel, and the main focus of the paper centers around solving the Bellman equation, which is a well-studied distributionally robust linear program extensively explored in the fields of optimization and mathematical programming. As a result, certain results claimed in the paper, such as Propositions 4.1 and 4.3, appear rather obvious. Additionally, it seems that some of the problems, such as those presented in Equations (11) and (13), bear a strong resemblance to those examined in a previous work cited as [1]. It is possible that similar techniques have been employed to derive the bisection algorithm and determine the running time complexities. In this regard, the technical contributions seem quite incremental.

Some theorems and corollaries are not properly stated. Please see the Question section.

The experiments conducted in the paper do not achieve a satisfactory level of validation. The comparisons made are limited to just two basic baselines, which fail to provide sufficient justification for the introduction of the bisection algorithm. For example, the paper suggests that (5)  and (8) can be solved directly using convex optimization, rather than employing bisection. Therefore, it would be necessary to include a comparison between solving (5) and (8) through convex optimization and the proposed approach.

[1] Chin Pang Ho, Marek Petrik, and Wolfram Wiesemann. Fast bellman updates for robust mdps. In International Conference on Machine Learning, pages 1979–1988. PMLR, 2018.

**Questions:**

- What would happen if you solve (5) and (8) by convex optimization? How does this affect the performance,  in terms of both theoretical complexities and numerical experiments?
- In Theorem 4.4 and Corollary 5.2.1, what is the quality of the returned solutions? Does the bisection return an optimal solution or just a near-optimal one? Does the quality of the returned solution depend on $\epsilon_1$ and $\epsilon_2$
- What are the technical distinctions between the proposed algorithm and the bisection method presented in [1]
- Can the results be extended to the robust MDP with (s)-rectangular ambiguity sets?

[1] Chin Pang Ho, Marek Petrik, and Wolfram Wiesemann. Fast bellman updates for robust mdps. In International Conference on Machine Learning, pages 1979–1988. PMLR, 2018.

**Limitations:**

I do not see any negative societal impact from this work.

---

> ### Author Rebuttal · Authors · 2023-08-07
>
> Thanks a lot for your encouraging comments and your time to review our paper!
>
> **Weaknesses**
>
> 1. The WDRMDP problem itself is not novel...well-studied distributionally robust linear program extensively explored in the fields of optimization and mathematical programming...Propositions 4.1 and 4.3, appear rather obvious...Equations (11) and (13), bear a strong resemblance to those examined in a previous work cited as [i]. It is possible that similar techniques have been employed...the technical contributions seem quite incremental.
>
> We would like to clarify that, although distributionally robust linear programs are extensively explored in the fields of optimization and mathematical programming, our contributions focus on the exploitation of the specific problem structure of distributionally robust Bellman updates. In particular, the proposed decomposition scheme does not work for generic distributionally robust linear programs.
>
> In the same spirit of using first order methods for solving both the robust MDPs (RMDPs) and the distributionally robust MDPs (DRMDPs) [ii-iii], while we agree that the proposed algorithm belongs to the class of bisection methods that solve RMDPs (such as [i]), we would like to highlight the differences between our work and [i]. (a) Firstly, we consider DRMDPs, which is a generalization of RMDPs, and so we are different from [i] in terms of problem settings. (b) Secondly, the resemblance of our bisection step and [i] only occurs in lines 200 - 213, which is less than 1/2 page in this paper. In particular, the second bisection step in the proposed nested bisection method and Proposition 4.3 are novel in this paper; without them, it is not easy to design algorithm(s) that has a time complexity that is almost linear in the number of expected kernels, which do not exist in RMDPs and [i]. In our opinion, this is one of the key factors that our proposed algorithm outperforms state-of-the-art algorithms, such as [iii]. (c) Finally, the subproblems, such as (11), are different from the case of RMDPs, and so it requires new theoretical results and algorithms for fast computations. Thank you for your question and we will clarify this point in our next version of the manuscript.
>
> 2. Some theorems and corollaries are not properly stated. Please see the Question section. Also, The experiments conducted in the paper do not achieve a satisfactory level of validation.
>
> Please find our answers to your questions below.
>
> **Questions**
>
> 1. What would happen if you solve (5) and (8) by convex optimization? How does this affect the performance, in terms of both theoretical complexities and numerical experiments?
>
> Sorry for the confusion caused. In our experiments, one of our baselines is Gurobi, which is a state-of-the-art commercial solver for solving convex programs, such as (5) and (8). In Section 6, we provide the comparisons between our proposed algorithms and Gurobi (i.e. convex optimization solver) on the runtimes of solving (5), and our algorithm outperforms Gurobi. In terms of solving (8), below please find our additional results for this rebuttal. As we can see, the results are consistent with our conclusion in Section 6.
>
> S=A=N=50, Gurobi (q=1): 722ms, fast (q=1): 6.10ms, Gurobi (q=2): 2959ms, fast (q=2): 32.61ms
>
> S=A=N=70, Gurobi (q=1): 998ms , fast (q=1): 6.26ms, Gurobi (q=2): 5372ms, fast (q=2): 35.21ms
>
> S=A=N=90, Gurobi (q=1): 1292ms, fast (q=1): 6.91ms, Gurobi (q=2): 8861ms, fast (q=2): 35.82ms
>
> In terms of theoretical complexities, from [iv-v], the time complexities for solving (5) using convex optimization solver are $\mathcal{O}(N^{4.5}S^{4.5}A^{4.5})$ and $\mathcal{O}(N^{3.5}S^{3.5}A^{3.5}\log \epsilon^{-1})$ for $q=1$ and $q=2$, respectively. The time complexities for solving (8) using convex optimization solver are $\mathcal{O}(N^{4.5}S^{4.5})$ and $\mathcal{O}(N^{3.5}S^{3.5}\log \epsilon^{-1})$ for $q=1$ and $q=2$, respectively. Therefore, the theoretical complexities of the proposed algorithms, as stated in Section 5, are orders of magnitudes lower than using generic convex optimization solvers.
>
> 2.  In Theorem 4.4 and Corollary 5.2.1, what is the quality of the returned solutions? Does the bisection return an optimal solution or just a near-optimal one? Does the quality of the returned solution depend on $\epsilon_1$ and $\epsilon_2$?
>
> The proposed algorithm is exact (up to tolerances $\epsilon_1$ and $\epsilon_2$). In particular, we have proved a new result on the error bound for the Bellman update, which is $\mathcal{O}(\epsilon_1+\epsilon_2)$. Since the complexities of the proposed algorithms are in $\mathcal{O}(\log\epsilon_1^{-1}\log\epsilon_2^{-1})$, one can compute a highly accurate solution with very small $\epsilon_1$ and $\epsilon_2$. We will provide this result in our next version of manuscript.
>
> 3. What are the technical distinctions between the proposed algorithm and the bisection method presented in [i].
>
> Please refer to the **Weaknesses**
>
> 4.  Can the results be extended to the robust MDP with (s)-rectangular ambiguity sets?
>
> Thank you for your insightful question.  The problem of interest, s-rectangular distributionally robust MDPs, is in fact a generalization of s-rectangular robust MDPs. Thus, distributionally robust MDPs, can recover (instead of extend) to s-rectangular robust MDPs. By setting the radius of the Wasserstein ball to be $\infty$ (or a large enough number, such as the diameter of $(\Delta_S)^A$), the ambiguity set contains all possible probability distributions. It is known that the worst-case distribution is a Dirac distribution, hence the distributionally robust MDP degenerates into robust MDP.
>
> [i] Fast Bellman updates for robust MDPs. 2018.
>
> [ii] Scalable first-order methods for robust MDPs. 2021.
>
> [iii] First-order methods for Wasserstein distributionally robust MDP. 2021.
>
> [iv] A new polynomial-time algorithm for linear programming. 1984.
>
> [v] Applications of second-order cone programming. 1998.

---

### Official Review · Reviewer_NtRj · 2023-07-07

**Soundness:** 3 good
**Presentation:** 3 good
**Contribution:** 3 good
**Rating:** 5
**Confidence:** 1

**Summary:**

The paper focuses on the computational complexity of Wasserstein Distributionally Robust MDPs with Lp norm. By decomposing the calculation of Bellman updates to smaller subproblems, the algorithm can achieve linear complexity in the number of actions and kernels, and quasi-linear complexity in the number of states.

**Strengths:**

The proposed method shows state-of-the-art complexity results.

**DISCLAIMER:**
 I have not checked the proof thoroughly and cannot verify the correctness of the theorems.

**Weaknesses:**

* The experiments are carried out on very simple, randomly generated distributionally robust MDP. While this approach provides a controlled setting for their work, it risks oversimplifying the problem and limiting the potential real-world applicability of the proposed methods.

**Questions:**

* The paper follows the common rectangularity assumption. However, it is known that the rectangularity assumption would lead to overly conservative policies. Can the proposed method easily extend to non-rectangular settings?
* How will the problem decomposition in section 4 affect the resulting value of the Bellman update? In other words, will there be tradeoffs between the computational complexity and the accuracy of the optimal value introduced by the proposed decomposition?
* Considering that the following paper also focuses on the model-based distributionally robust RL setting. It would be interesting to also compare with it.
    * Shi, Laixi, and Yuejie Chi. "Distributionally robust model-based offline reinforcement learning with near-optimal sample complexity." arXiv preprint arXiv:2208.05767 (2022).


**Limitations:**

The limitation of the proposed method is not fully discussed in the paper.

---

> ### Author Rebuttal · Authors · 2023-08-07
>
> Thank you very much for your comments and your time to review our paper!
>
> **Weaknesses**
>
> 1. The experiments are carried out on very simple, randomly generated distributionally robust MDP. While this approach provides a controlled setting for their work, it risks oversimplifying the problem and limiting the potential real-world applicability of the proposed methods.
>
> We are sorry for being unclear on the purpose of our experimental setup. This paper focuses on algorithmic development and computational complexity for solving distributionally robust MDPs. Therefore, randomly generated distributionally robust MDPs are perfect for our purpose, as additional problem-specific structures should not be considered for the sake of generality. Moreover, these randomly generated distributionally robust MDPs are more computationally challenging than stylized problems as their transitional kernels are dense, and so they are used to test the performance of various algorithms in our experiments.
>
> **Questions**
>
> 1. The paper follows the common rectangularity assumption. However, it is known that the rectangularity assumption would lead to overly conservative policies. Can the proposed method easily extend to non-rectangular settings?
>
> Thank you for raising this interesting question. Solving the general robust MDP problem with nonrectangular ambiguity sets is strongly NP-hard and intractable [ii]. Moreover, Bellman equations for nonrectangular robust MDPs do not exist in general, so our approach could not be extended to non-rectangular settings. However, the rectangularity assumption can always be satisfied via outer approximation [i]. Therefore, most research focuses on rectangular robust MDPs [i-iii], such as $s$-rectangular and $s,a$-rectangular robust MDPs, and so does the research in distributionally robust MDPs [iv-v]. In this paper, we adopt the $s$-rectangular setting, which is less conservative than the $s,a$-rectangular case.
>
> 2. How will the problem decomposition in section 4 affect the resulting value of the Bellman update? In other words, will there be tradeoffs between the computational complexity and the accuracy of the optimal value introduced by the proposed decomposition?
>
> We are sorry for the confusion caused. By exploiting the specific problem structure of the optimization problem for the distributionally robust Bellman update, the proposed algorithm is exact (up to user-specified tolerances on the accuracy). Therefore, we can compute the Bellman update exactly up to an arbitrarily small tolerance. We will clarify this point in our next version of manuscript.
>
> 3. Considering that the following paper also focuses on the model-based distributionally robust RL setting. It would be interesting to also compare with it.
> Shi, Laixi, and Yuejie Chi. "Distributionally robust model-based offline reinforcement learning with near-optimal sample complexity." arXiv preprint arXiv:2208.05767 (2022).
>
> Thank you for bringing up this interesting reference paper. We would like to clarify that our paper focuses on the computations and time complexity of distributionally robust MDPs, but the reference paper focuses on learning and sample complexity. Therefore, the purposes of both papers are complementary to each other. Moreover, by our definition in Section 1 (which is also adopted in [i-v]), this reference paper considers the settings of robust MDPs rather than distributionally robust MDPs; thus, the problem settings are different in the two papers. We are sorry for the confusion caused, and we will cite this paper and clarify this point in our next version of manuscript.
>
> **Limitations**
>
> 1. The limitation of the proposed method is not fully discussed in the paper.
>
> We apologize for missing this discussion. Our method does not consider the settings of continuous state and action spaces. Besides, we focus on discounted MDPs, and it is not obvious that our proposed algorithms could be extended to the case of optimizing average expected return. We will provide the discussion in the next version of manuscript.
>
>
>
>
>
> [i] A. Nilim and L. El Ghaoui, Robust control of Markov decision processes with uncertain transition matrices, 2005.
>
> [ii] W. Wiesemann, D. Kuhn, and B. Rustem, Robust Markov decision processes, 2013.
>
> [iii] G. N. Iyengar, Robust dynamic programming, 2005.
>
> [iv] I. Yang, A convex optimization approach to distributionally robust Markov decision processes with Wasserstein distance, 2017.
>
> [v] J. Grand-Clement and C. Kroer, First-order methods for Wasserstein distributionally robust MDP, 2021.
>
> [vi] L. Shi and Y. Chi, Distributionally robust model-based offline reinforcement learning with near-optimal sample complexity, 2022.

---

> > ### Comment · Reviewer_NtRj · 2023-08-14
> > **Re**
> >
> > Thanks for the clarification. I raised my rating to 5.

---

> > > ### Author Response · Authors · 2023-08-14
> > > **Thank you!!**
> > >
> > > Thank you very much for reading our rebuttal, recognizing our work, and updating our score!

---

### Decision · Program_Chairs · 2023-09-21

**Decision:**

Accept (poster)

**Comment:**

Robust MDP (RMDP) and robust RL are now a popular approach for modeling and solving robust control problems. While the basic formulation is straight forward, it typically hides the complexity of the inner optimization over the uncertainty set. In fact, many robust RL algorithms are not scalable to large state/action spaces due to the complexity of this optimization. This paper presents a computationally efficient solution framework for solving distributionally robust MDPs with Wasserstein ambiguity sets.

I have read this paper, reviewer comments, and the authors’ rebuttal carefully. The paper is well written and technically solid. The contribution is novel and will be of interest and use to the robust RL community. Reviewers had some questions about the experiments, and I the authors have given clear response those.

For the final version, please make all the changes you have promised during the rebuttal, and update the paper.

Regards,

AC